# PROGRESSIVE GRAPH STRUCTURE ADJUSTMENT FOR HOMOPHILY SHIFT IN GRAPH DOMAIN ADAPTATION

## ABSTRACT

Node homophily shift—the mismatch in the tendency of nodes to have neighbors with the same label between source and target graphs—poses a key challenge for *Graph Domain Adaptation* (*GDA*) without target labels. We introduce *Progressive Structure Adjustment for Homophily Shift* (*PSAHS*), which progressively reduces homophily discrepancies: in the source graph by modifying existing edges and adding new edges for low-homophily nodes, and in the target graph by making analogous adjustments for nodes with consistent label predictions from *Graph Neural Networks* (*GNNs*) and *Multi-Layer Perceptrons* (*MLPs*). After each refinement, GNNs are updated with domain-adversarial training for representation alignment. This interplay of structure adjustment and representation learning mitigates homophily shift, tightens the target error bound, and yields consistent improvements over strong baselines, highlighting the necessity of node homophily alignment for effective cross-graph transfer.

## 1 INTRODUCTION

*Graph Neural Networks* (*GNNs*) have achieved remarkable success in node classification by jointly leveraging node attributes and graph structure. However, in many real-world applications—such as cross-network recommendation (Zhao et al., 2025), bioinformatics (Li et al., 2025), and citation analysis (He et al., 2023)—target-domain labels are scarce or even entirely unavailable, making it difficult to train reliable models directly. This challenge motivates the study of *Graph Domain Adaptation* (*GDA*), which aims to transfer knowledge from a well-labeled source graph to an unlabeled or sparsely labeled target graph.

The central difficulty in GDA lies in distributional shifts between source and target domains, spanning node attributes, graph structures, and label distributions. Existing GDA methods primarily address this issue by aligning node features through adversarial training (Zhang et al., 2019; Wu et al., 2020) or direct feature alignment (Wu et al., 2023; Chen et al., 2025), and then training a shared classifier in the aligned space. However, since these approaches can also be applied to non-graph data, they often overlook the unique structural properties of graphs and fail to capture how graph-specific structures affect label prediction.

Recent work has started to explore structural shifts conditioned on labels. For example, Liu et al. (2023; 2024c) model that the probability of an edge between two nodes is determined by the labels of the node pair. To mitigate the class-conditional structure shift, they reweight each edge by the ratio of class-conditional edge probabilities between domains, estimated from the labels of two endpoints. Therefore, their shift formulation only captures the structure–label relation at the level of a *single edge* level, making their approaches remain inherently local: each edge is adjusted solely based on its *two endpoints'* labels, without accounting for broader or global patterns of structural mismatch.

This limitation highlights the central importance of *node homophily* (Mao et al., 2023)—the proportion of a node's neighbors that share its label—which reflects the ability of an ego-network to propagate label information. Node homophily thus captures a global structure–label property that goes beyond isolated edges. Most prior work on homophily focuses on increasing the overall graph-level homophily ratio (Zhu et al., 2020) by incorporating higher-order neighbors with the same labels (Li et al., 2022) and reconstructing the graph via spectral clustering (Li et al., 2023). More detailed related work is listed in Appendix A.2. By contrast, variation in node-level homophily has received much less attention until studies (Ma et al., 2021; Mao et al., 2023), which show that within homophilic graphs, GNNs perform well on high-homophily nodes but deteriorate on low-homophily

ones, and that such node homophily shifts strongly affect generalization in the node classification problem. In the GDA setting, Fang et al. (2025b) further demonstrates that mismatched homophily distributions—i.e., differences in the composition of high- versus low-homophily nodes across domains—can significantly hinder knowledge transfer even when feature distributions are aligned. Yet they still treat node homophily as intrinsic and immutable, attempting to address it only indirectly through feature alignment rather than as a structure-label relation shift that can be explicitly reduced.

We propose *Progressive Structure Adjustment for Homophily Shift (PSAHS)*, a new paradigm for GDA grounded in a generalization error bound that links target performance to source error, homophily shift, and representation divergence. Our three-stage framework first enhances source homophily by modifying the adjacency matrix entries of low-homophily nodes, then refines the target graph by adjusting low-homophily nodes with consistent label predictions from the GNN and MLP to align its homophily distribution with the source, and finally updates the GNN encoder via domain-adversarial training for representation alignment. By alternating target refinement and representation alignment, PSAHS progressively mitigates homophily shift, improves the reliability of label prediction, and forms a self-reinforcing loop between structure adjustment and representation learning. Guided by source labels and reliable target label predictions, PSAHS explicitly raises low node homophily, provably reduces homophily shift, and enhances GDA performance.

Our contributions are threefold: *(i)* We present a theoretical analysis that explicitly connects homophily shift to cross-domain generalization error, motivating structural adjustment as a principled solution. *(ii)* We propose a progressive homophily-aware structure adjustment framework that alternates between target graph refinement and representation alignment. *(iii)* We validate our method on multiple benchmarks, showing consistent improvements over strong GDA baselines, with especially large gains under severe homophily mismatch, underscoring the importance of structural alignment for effective cross-graph transfer.

## 2 PRELIMINARIES

### 2.1 NODE CLASSIFICATION

A graph is represented as $\mathcal{G} = (\mathcal{V}, \mathcal{E}, X)$, where $\mathcal{V}$ is the node set with $|\mathcal{V}| = n$, $\mathcal{E}$ is the edge set, and $X := (X_u)_{u \in \mathcal{V}} \in \mathbb{R}^{n \times F}$ is the node attribute matrix, with each row $X_u \in \mathbb{R}^F$ denoting the attribute vector of node $u$. The adjacency matrix $A = (A_{uv})_{u,v \in \mathcal{V}}$ encodes the graph structure, where $A_{uv} \in \{0, 1\}$ indicates whether an edge exists between nodes $u$ and $v$. For any $n \in \mathbb{N}^*$, we denote $[n] := \{1, 2, \ldots, n\}$. We focus on the node-level classification task, where the goal is to predict the label vector $Y := (Y_u)_{u \in \mathcal{V}}$. For theoretical clarity, we assume binary labels $Y_u \in \mathcal{Y} := \{0, 1\}$, though our approach can be naturally extended to multi-class settings.

*Graph Neural Networks* (*GNNs*) (Wu et al., 2021) have become the dominant framework for graph representation learning. An $(L-1)$-layer GNN iteratively updates node representations via message passing. Formally, given adjacency matrix $A$ and attribute matrix $X$, the GNN produces features $f := \phi(X, A) \in \mathbb{R}^{n \times F}$ through $L - 1$ propagation layers. The final feature $f$ is then fed into a classifier $g : \mathbb{R}^{n \times F} \to \mathbb{R}^{n \times M}$, where $M$ is the number of classes. The overall model is $g \circ \phi(X, A)$, and the output $g_{u,m}(\phi(X, A))$ gives the predicted probability that node $u$ belongs to class $m$.

### 2.2 GRAPH DOMAIN ADAPTATION AND NODE HOMOPHILY SHIFT

*Graph Domain Adaptation* (*GDA*) studies the problem of transferring knowledge from a labeled source graph to an unlabeled target graph, where the two domains exhibit distributional differences. Formally, the source domain provides a labeled graph $\mathcal{G}_S = (\mathcal{V}_S, \mathcal{E}_S, X^S)$ with $(X^S, A^S, Y^S) \sim P_S$, while the target domain provides an unlabeled graph $\mathcal{G}_T = (\mathcal{V}_T, \mathcal{E}_T, X^T)$ with $(X^T, A^T) \sim P_T$. The distributions $P_S$ and $P_T$ may differ in node attributes, graph topology, and even conditional label distributions. The goal of GDA is to minimize the classification risk on $P_T$ by leveraging labeled data from $P_S$ while accounting for these shifts.

A central structural property of graphs is *node homophily*, which measures the tendency of connected nodes to share the same label. For a node $u \in \mathcal{V}$ with neighborhood $\mathcal{N}_u := \{v \in \mathcal{V} \mid A_{uv} = 1\}$, the *homophily ratio* is defined as $h_{\mathcal{G}}(u) := \frac{1}{d_u} \sum_{v \in \mathcal{N}_u} \mathbf{1}\{Y_u = Y_v\}$, where $\mathbf{1}\{\cdot\}$ is the indicator function and $d_u := |\mathcal{N}_u|$ is the degree of node $u$. This ratio captures the proportion of neighbors of $u$ that share its label. For notational clarity, we denote the homophily ratios in the source and target

graphs as $h_S(u) := h_{\mathcal{G}_S}(u)$ and $h_T(u) := h_{\mathcal{G}_T}(u)$, respectively. The collection of all node-level ratios $\{h_{\mathcal{G}}(u) : u \in \mathcal{V}\}$ defines the *homophily distribution* $P_{\mathcal{G}}(h)$ of a graph $\mathcal{G}$.

**Definition 2.1** (Node Homophily Shift). Let $P_S(h)$ and $P_T(h)$ denote the node homophily distributions of the source and target graphs, respectively. We define a *node homophily shift* as the case where $P_S(h) \neq P_T(h)$.

In this work, we study GDA by focusing on reducing the *node homophily shift* between the source and target domains. This shift has been empirically observed across a wide range of citation and social networks in Fang et al. (2025b). Our subsequent theoretical analysis shows that the target-domain error bound explicitly depends on the node homophily shift, highlighting its impact on cross-domain generalization.

| Symbol | Meaning | Symbol | Meaning |
|---|---|---|---|
| $\mathcal{G} = (\mathcal{V}, \mathcal{E}, X)$ | graph with nodes, edges, and attributes | $\widetilde{A}^S, \widetilde{A}^T$ | adjusted adjacency matrix of source/target domain |
| $X^S, X^T$ | node feature matrix of source/target domain | $\widetilde{f} := \phi(X, \widetilde{A})$ | GNN encoded feature matrix with adjusted structure |
| $A^S, A^T$ | adjacency matrix of source/target domain | $\widetilde{h}_S(u), \widetilde{h}_T(u)$ | node homophily in adjusted source/target graph |
| $Y^S$ | node labels of the source domain | $\widehat{Y}_u$ | the pseudo-label of node $u$ |
| $\phi, g$ | GNN encoder, GNN classifier | $\widehat{h}_T(u)$ | the estimated node homophily |
| $f := \phi(X, A)$ | GNN encoded feature matrix, | $h$ | the desired node homphily threshold |
| $g_{u,m}(\phi(X, A))$ | predicted prob. of class $m$ for node $u$, | $\alpha_u$ | the node-wise edge adjustment strength |
| $h_S(u), h_T(u)$ | node homophily in source/target graph | $\mathcal{V}_T^r$ | reliable target node set where the GNN and MLP agree |

Table 1: Summary of key notations used in the paper.

# 3 THEORETICAL ANALYSIS OF STRUCTURE ADJUSMENT STRATEGY UNDER NODE HOMOPHILY SHIFT

To mitigate the homophily shift, a fundamental approach is to directly adjust node homophily in both domains, thereby reducing the homophily gap. Since a node's homophily ratio is jointly determined by the graph structure and node labels, a natural strategy is to manipulate the graph structure rather than labels to mitigate the node homophily shift. Adjusting labels would inevitably introduce noise and degrade classifier reliability, whereas structural refinement provides a principled way to modify homophily while preserving label consistency. Formally, let $\widetilde{A}^S$ and $\widetilde{A}^T$ denote the adjusted adjacency matrices of the original $A^S$ and $A^T$ in the source and target domains, respectively. By definition, $\widetilde{A}^S$ and $A^S$ have the same dimensions, but there exist nodes $u, v \in \mathcal{V}^2$ such that $\widetilde{A}_{uv}^S \neq A_{uv}^S$. An analogous definition applies to $\widetilde{A}^T$ and $A^T$. Given $\widetilde{A}^S$ and $\widetilde{A}^T$, we denote the corresponding homophily ratio and aggregated feature of node $u$ as $\widetilde{h}_u$ and $\widetilde{f}_u$, respectively.

After transforming from the original $A^S$ and $A^T$ to the adjusted structure $\widetilde{A}^S$ and $\widetilde{A}^T$, we study a GNN classifier trained on the adjusted source graph and applied to the adjusted target graph. In what follows, we derive a target-domain error bound for this classifier, which provide explicit guidance for designing a specific structure adjustment approach and lays the theoretical foundation for our subsequent methodology.

Following Mao et al. (2023), we adopt the *Simplifying Graph Convolutional Networks* (*SGN*) model (Wu et al., 2019) as the base GNN classifier. In this setting, the classifier $g \circ \phi$ is an MLP operating on aggregated features, formally defined as $g \circ \phi(X, A) := \text{MLP}\big(D^{-1}AX; \{W^\ell\}_{\ell=1}^L\big)$, where $D$ is the degree matrix and $W^\ell$ are the learnable parameters of the $\ell$-th layer. For theoretical analysis, we consider the *margin loss function* with margin parameter $\gamma \geq 0$: $\widehat{\mathcal{R}}_S^\gamma(g \circ \phi) := \frac{1}{n_s} \sum_{i=1}^{n_s} \mathbf{1}\{g_{i,Y^S}(\phi(X^S, A^S)) \leq \gamma + \max_{k \neq Y^S} g_{i,k}(\phi(X^S, A^S))\}$. The expected margin loss is then $\mathcal{R}_S^\gamma(g \circ \phi) := \mathbb{E}_{Y_u \sim P_S(Y|f_u(X^S, \widetilde{A}^S))}\big[\widehat{\mathcal{R}}_S^\gamma(g \circ \phi)\big]$. When $\gamma = 0$, this reduces to the standard classification loss $\mathcal{R}_S(g \circ \phi) := \mathcal{R}_S^0(g \circ \phi)$. Similar definitions hold for the target domain.

Building on the PAC-Bayesian framework, we next derive theoretical results showing how graph structure adjustment influences target-domain error through the adjusted homophily ratios and aggregated features in the two domains.

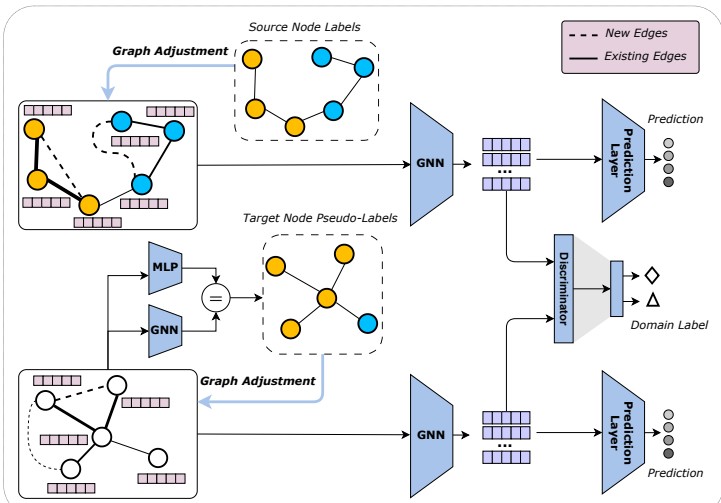

Figure 1: Framework of Progressive Structure Adjustment for Homophily Shift (PSAHS).

**Theorem 3.1.** *Under the SGN model, let $g \circ \phi$ be a classifier from the hypothesis space of GNN classifiers. Denote by $\widetilde{f}_u$ (resp. $\widetilde{f}_v$) the aggregated feature of a source node $u \in \mathcal{V}_S$ (resp. target node $v \in \mathcal{V}_T$) under the adjusted graph structure $\widetilde{A}^S$ (resp. $\widetilde{A}^T$). Similarly, let $\widetilde{h}_S(u)$ and $\widetilde{h}_T(v)$ denote the corresponding node homophily ratios. Then for any $\gamma > 0$, $\delta \in (0, 1)$, $\alpha \in (0, 1/4)$, and sufficiently large $n_s$, there exists a constant $c$ independent of $n$ such that with probability at least $1 - \delta$, the target margin loss $\mathcal{R}_T(g \circ \phi)$, can be upper bounded by*

$$\widehat{\mathcal{R}}_S^\gamma(g \circ \phi) + c\left(\frac{1}{n_s n_t} \sum_{u \in \mathcal{V}_S} \sum_{v \in \mathcal{V}_T} \left(|\widetilde{h}_S(u) - \widetilde{h}_T(v)| + \|\widetilde{f}_u - \widetilde{f}_v\|_2\right) + \frac{1}{n_s^\alpha} + \frac{\ln(1/\delta)}{n_s^{2\alpha}}\right). \quad (1)$$

Therefore, Theorem 3.1, whose proof is given in Appendix C.1, establishes that minimizing the target classification error requires jointly reducing the following three components.

**(I) Empirical source margin loss $\widehat{\mathcal{R}}_S(g \circ \phi)$.** This term reflects the classification performance on the source domain. As shown in Mao et al. (2023), since homophily shifts between high- and low-homophily node subgroups degrade performance on the minority subgroup, making it difficult for GNNs to perform well on both simultaneously. Hence, the source error can be reduced by increasing the node homophily of low-homophily nodes. To this end, we refine the source graph structure by down-weighting inter-class edges and introducing additional intra-class edges for low-homophily nodes, as described in Section 4.1.

**(II) Discrepancy in node homophily ratios across domains $\sum_{u \in \mathcal{V}_S} \sum_{v \in \mathcal{V}_T} |\widetilde{h}_S(u) - \widetilde{h}_T(v)|$.** After adjusting the source graph as in *(I)*, the values $\widetilde{h}_S(u)$ become fixed. To reduce the remaining discrepency, we adjust the target graph structure to modify $\widetilde{h}_T(v)$, thereby aligning the homophily distributions of the two domains (see Section 4.2).

**(III) Discrepency in aggregated node representations $\sum_{u \in \mathcal{V}_S} \sum_{v \in \mathcal{V}_T} \|\widetilde{f}_u - \widetilde{f}_v\|_2$.** This term quantifies representation-level misalignment between domains. We address it by aligning the distributions of node representations through domain-adversarial neural networks (see Section 4.3).

It is important to note that prior work such as Fang et al. (2025b) also introduces a homophily shift term similar to *(II)*. However, in their formulation, the homophily ratios $h_S(u)$ and $h_T(v)$ are intrinsic and fixed by the original graph structure, making them unmodifiable. In contrast, our framework leverages the adjusted homophily ratios $\widetilde{h}_S(u)$ and $\widetilde{h}_T(v)$, which can be actively refined through structural adjustments. This flexibility allows us not only to enhance source homophily (thereby reducing *(I)*) but also to explicitly align homophily distributions across domains (thereby reducing *(II)*), jointly tightening the target error bound.

# 4 METHODOLOGY

In this section, we propose a homophily-aware structure adjustment framework for graph domain adaptation to mitigate the three error components shown in Theorem 3.1. As shown in Figure 1, our method progressively refines graph structures in both domains while aligning node representations, thus minimizing the three error components in the error bound. Specifically, we *(i)* enhance source homophily by modifying inter-class edges and adding intra-class edges for low-homophily nodes, then train a GNN and MLP to generate target pseudo-labels; *(ii)* reduce cross-domain homophily shift by adjusting target structures based on the consistent pseudo-labels; and *(iii)* mitigate representation misalignment via domain-adversarial training. These steps alternate iteratively until convergence, producing an effective target-domain classifier with tighter error bounds.

## 4.1 ENHANCING SOURCE HOMOPHILY VIA GRAPH STRUCTURE ADJUSTMENT

To improve classification performance in the source domain, our goal is to increase node homophily ratios, particularly for nodes with initially low homophily, as motivated by the theoretical results in Appendix B.

We refine the source graph structure by reweighting edges of low-homophily nodes and introducing additional intra-class connections, while keeping the adjacency of high-homophily nodes unchanged. Specifically, for each node $u$ with $h_S(u) < h$, we decrease the weights of its inter-class edges to $1 - \alpha_u$ and increase the weights of its intra-class edges to $1 + \alpha_u$, where $\alpha_u \in [0, 1]$ is a node-specific edge adjustment strength, with its precise value provided in Theorem 4.2. To further promote intra-class connectivity, we randomly select $d_u(1 - h_S(u))$ non-neighbor nodes $v$ with $Y_v = Y_u$ and connect them to $u$ with weight $\alpha_u$. For nodes with $h_S(u) \geq h$, we retain their original adjacency entries. Formally, the adjusted adjacency matrix $\widetilde{A}^S$ is defined as

$$\widetilde{A}_{uv}^S := \begin{cases} A_{uv}^S, & \text{if } v \in \mathcal{V}_S, \ h_S(u) \geq h, \\ 1 + \alpha_u, & \text{if } v \in \mathcal{V}_S, \ h_S(u) < h, \ A_{uv}^S = 1, \ Y_u = Y_v, \\ 1 - \alpha_u, & \text{if } v \in \mathcal{V}_S, \ h_S(u) < h, \ A_{uv}^S = 1, \ Y_u \neq Y_v, \\ \alpha_u, & \text{if } v \in \mathcal{V}_S, \ h_S(u) < h, \ A_{uv}^S = 0, \ v \in \mathcal{N}_u', \\ 0, & \text{if } v \in \mathcal{V}_S, \ h_S(u) < h, \ A_{uv}^S = 0, \ v \notin \mathcal{N}_u', \end{cases} \tag{2}$$

where $\mathcal{N}_u' \subset \mathcal{V} \setminus \mathcal{N}_u$ denotes the newly added set of same-label neighbors for node $u$.

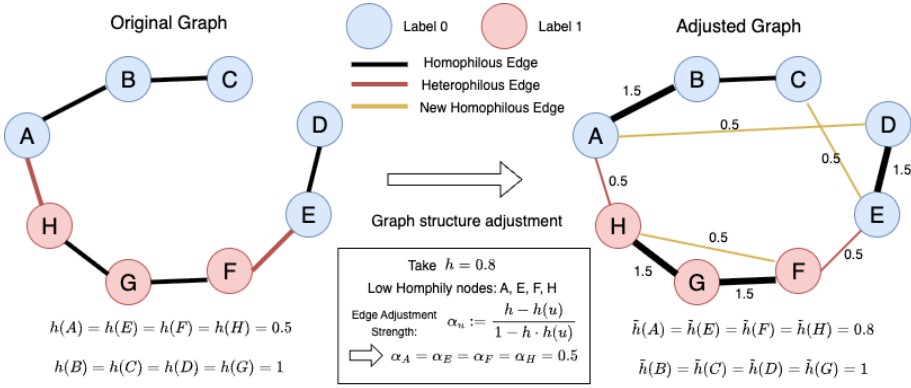

Figure 2: Left subfigure: four nodes A,B,C,D,E from class 0 and three nodes F,G,H from class 1. The homophily ratios of nodes A,E,F,H equal 1, while those of nodes B,C,D,G equal 0.5. Middle box: Set the desired homophily level to $h = 0.8$, nodes B,C,D,G are identified as low-homophily nodes. For each $u \in \{B, C, D, G\}$, the edge adjustment strength is computed as $\alpha_u = h - h(u)/(1 - h \cdot h(u)) = (0.8 - 0.5)/(1 - 0.8 \cdot 0.5) = 0.5$. Right subfigure: for each low-homophily node $u \in \{B, C, D, G\}$, we increase its homophilous edge weight to $1 + \alpha_u = 1.5$ (thicker black edges), decrease its heterophilous edge weight to $1 - \alpha_u = 0.5$ (thinner green edges), and add new homophilous edges with the strength $\alpha_u = 0.5$ (thin yellow edges). As a result, the adjusted homophily ratios defined in Eq. (3) increase to $\widetilde{h}(u) = 0.8$. This example visually demonstrates how our strategy ensures that every node reaches the desired homophily level $h$.

Since $\widetilde{A}^S$ is no longer binary, we extend the definition of the homophily ratio under the adjusted structure as

$$\widetilde{h}_S(u) = \frac{\sum_{v \in \mathcal{V}_S,\, Y_u = Y_v} \widetilde{A}^S_{uv}}{\sum_{v \in \mathcal{V}_S} \widetilde{A}^S_{uv}}. \tag{3}$$

For nodes with $h_S(u) \geq h$, the ratios remain unchanged, i.e., $\widetilde{h}_S(u) = h_S(u) \geq h$. For nodes with $h_S(u) < h$, the following theorem guarantees that the adjustment increases their homophily to at least the desired threshold:

**Theorem 4.1.** *For any $h \in (0, 1]$, if $\alpha_u \in \big[(h - h_S(u))/(1 - h \cdot h_S(u)), 1\big)$ is chosen in Eq. (2) for a node $u$ with $h_S(u) < h$, then the adjusted homophily in Eq. (3) satisfies $\widetilde{h}_S(u) \geq h$.*

Theorem 4.1, whose proof is given in Appendix C.2, demonstrates that our adjustment strategy successfully elevates the homophily of initially low-homophily nodes above the specified threshold, thereby enhancing source-domain classification and reducing the source margin loss term *(I)* in the target error bound in Eq. (10).

## 4.2 Aligning Target Homophily via Graph Structure Adjustment

We now aim to align node homophily ratios across domains to reduce the shift term *(II)* in Theorem 3.1. Recall that in the source domain, all nodes have been refined to achieve homophily at least $h$ through the adjustment in Eq. (2). Thus, the remaining task is to promote low-homophily nodes in the target domain to reach the same threshold $h$, thereby aligning homophily distributions across domains, which is guaranteed by Theorem 4.2. A key challenge is that computing homophily ratios requires node labels, which are unavailable in the target domain. To overcome this, we employ a GNN classifier $g \circ \phi$ trained on the source domain to generate pseudo-labels for target nodes: $\widehat{Y}_u := \arg\max_{m \in [M]} g_{u,m}(\phi(X^T, \widetilde{A}^T))$, $u \in \mathcal{V}_T$, where initially $\widetilde{A}^T := A^T$.

To improve the reliability of label prediction, we introduce an auxiliary MLP trained only on source data. We then identify target nodes where the GNN and MLP predictions agree as *reliable nodes*, which form the reliable target set $\mathcal{V}^r_T$. For high-homophily reliable nodes, GNNs are typically more accurate than MLPs, as homophilous edges enable the aggregation of more same-class features that enhance discriminability. For low-homophily nodes, MLPs often outperform GNNs, as MLPs are unaffected by the noisy signals introduced by heterophilous edges. Thus, when both models yield the same prediction—consistent from raw attributes $X$ and from aggregated features via the adjacency matrix—the prediction is regarded as high-confidence and reliable.

For each node in the reliable target set $u \in \mathcal{V}^r_T$, based on the target reliable pseudo-labels $\widehat{Y}_u$, we estimate its homophily ratio as

$$\widehat{h}_T(u) := \frac{\sum_{v \in \mathcal{N}_u \cap \mathcal{V}^r_T} \mathbf{1}\{\widehat{Y}_u = \widehat{Y}_v\}}{|\mathcal{N}_u \cap \mathcal{V}^r_T|}. \tag{4}$$

To mitigate the node homophily shift, we need to improve the homophily of reliable target nodes to the same threshold $h$ as used in the source domain. Reliable low-homophily nodes $u \in \mathcal{V}^r_T$ satisfying $\widehat{h}_T(u) < h$ are the target nodes whose adjacency entries $A_{u\cdot}$ are adjusted following a scheme similar to Section 4.1, while the adjacency entries of non-reliable nodes and reliable high-homophily nodes remain unchanged. Specifically, given the target labels are unobserved, the target adjacency matrix will be adjusted to $\widetilde{A}^T_{uv}$ in the same way as in Eq. (2) by only replacing $\mathcal{V}_S$, $Y_u$, $Y_v$ $h_S$ with $\mathcal{V}^r_T$, $\widehat{Y}_u$, $\widehat{Y}_v$, $\widehat{h}_T$, and changing the construction way of the newly added neighbor set $\mathcal{N}'_u$. To construct $\mathcal{N}'_u$ in the target domain, we select $(h - \widehat{h}_T(u))d_u$ reliable non-neighbor nodes $v \in \mathcal{V}^r_T \setminus \mathcal{N}_u$ *(i)* sharing the same predicted label $\widehat{Y}_v = \widehat{Y}_u$ and *(ii)* having the highest GNN confidence scores $g_{v,\widehat{Y}_u}(\phi(X^T, \widetilde{A}^T))$.

This edge refinement increases homophily for the adjusted low-homophily nodes and their neighbors, including originally heterophilous neighbors and newly connected same-class neighbors, thereby improving separability of the aggregated features. In particular, the neighbors of reliable

low-homophily nodes are often themselves low-homophily and prone to misclassification. Therefore, adjusting the edge weights between them is crucial for these neighbors to enhance node homophily and improve the predictive ability. Furthermore, by restricting edge adjustments only to low-homophily nodes with consistent predictions, our approach captures the key to performance improvement while avoiding erroneous adjustments.

The generalized homophily ratio of target node $u \in \mathcal{V}_T^r$ under the adjusted structure is then

$$\widetilde{h}_T(u) = \frac{\sum_{v \in \mathcal{V}_T^r, \widehat{Y}_u = \widehat{Y}_v} \widetilde{A}_{uv}^T}{\sum_{v \in \mathcal{V}_T^r} \widetilde{A}_{uv}^T}. \tag{5}$$

**Theorem 4.2.** *Let $\widetilde{h}_S$ and $\widetilde{h}_T$ be the adjusted homophily as in Eq. (3) and Eq. (5), respectively. For any $h \in (0, 1]$, if we take $\alpha_u = (h - h_S(u))/(1 - h \cdot h_S(u))$ for source nodes $u$ with $h_S(u) < h$ and $\alpha_v = (h - h_T(v))/(1 - h \cdot h_T(v))$ for target nodes $v$ with $h_T(v) < h$, then*

$$\sum_{u \in \mathcal{V}_S} \sum_{v \in \mathcal{V}_T} |\widetilde{h}_S(u) - \widetilde{h}_T(v)| \leq \sum_{u \in \mathcal{V}_S} \sum_{v \in \mathcal{V}_T} |h_S(u) - h_T(v)|.$$

*In other words, the shift in homophily ratios between domains after structure adjustment is no larger than that without adjustment.*

Theorem 4.2, whose proof is given in Appendix C.2, shows that, with proper node-wise choices of $\alpha_u$, adjusting the source and target graph structure as in the scheme of Eq. (2), can reduce the node homophily shift between domains. This result informs the selection of $\alpha_u$ values in our experiments. In summary, our adjustment strategy modifies existing edges and adds new intra-class edges based on reliable pseudo-labels. Unlike prior methods such as Liu et al. (2023; 2024c), which reweight all existing edges for all source nodes, our approach specifically targets low-homophily nodes in both domains and additionally introduces new homophilous edges, making the process more focused, adaptive, and effective for cross-domain alignment.

### 4.3 REPRESENTATION ALIGNMENT ACROSS DOMAINS

To address the discrepancy in aggregated features across domains and reduce the error term *(III)* in Theorem 3.1, we adopt a domain-adversarial training framework to learn a domain-invariant GNN encoder $\phi$. Specifically, we solve the following minimax problem:

$$\min_{\phi} \max_{\xi} \left[ \frac{1}{|\mathcal{V}_S|} \sum_{u \in \mathcal{V}_S} \log \left( \xi(\phi_u(X^S, \widetilde{A}^S)) \right) + \frac{1}{|\mathcal{V}_T|} \sum_{u \in \mathcal{V}_T} \log \left( 1 - \xi(\phi_u(X^T, \widetilde{A}^T)) \right) \right],$$

where $\xi$ is a domain discriminator. We denote the corresponding alignment loss as $\mathcal{R}_{\text{RA}}(\phi)$.

For supervised learning on the source domain, we use the cross-entropy loss:

$$\mathcal{R}_{\text{CE}}(\phi, g) = -\frac{1}{|\mathcal{V}_S|} \sum_{u \in \mathcal{V}_S} \mathcal{L}_{\text{CE}} \left( g_u(\phi(X^S, \widetilde{A}^S)), Y_u \right),$$

where $g_u(\cdot)$ denotes the predicted class probability for node $u$.

The overall training objective integrates representation alignment and source supervision:

$$\min_{\phi, g} \Big[ \underbrace{\mathcal{R}_{\text{CE}}(\phi, g)}_{\text{Supervised Loss}} + \underbrace{\gamma_{\text{RA}} \cdot \mathcal{R}_{\text{RA}}(\phi)}_{\text{Repres. Alignment}} \Big], \tag{6}$$

where $\gamma_{\text{RA}} > 0$ is a balancing hyperparameter. This adversarial framework encourages $\phi$ to generate domain-invariant representations while maintaining predictive power on the source.

### 4.4 PROGRESSIVE HOMOPHILY AND REPRESENTATION ALIGNMENT ACROSS DOMAINS

Our algorithm begins by fixing the adjusted source graph structure and training an initial GNN classifier using the labeled source data in Section 4.1. Then, it iteratively performs two interdependent steps: *(i)* adjusting edges in the target graph (Section 4.2) and *(ii)* updating the GNN parameters (Section 4.3). This progressive training scheme enables the target graph refinement and representation alignment to mutually reinforce each other, gradually enhancing target-domain performance and ultimately yielding a GNN classifier that minimizes the target error bound. The complete procedure is summarized in Algorithm 1.

---

**Algorithm 1** Progressive Structure Adjustment for Homophily Shift (PSAHS)

---

**Input:** Source graph $\mathcal{G}_S$ with labels $\mathcal{Y}_S$; unlabeled target graph $\mathcal{G}_T$; GNN encoder $\phi$ and classifier $g$; auxiliary MLP classifier; homophily threshold $h \in (0, 1)$.

    Adjust source graph adjacency to obtain $\widetilde{A}^S$ using Eq. (2).

    Train initial GNN classifier $g \circ \phi$ and auxiliary MLP on source data.

    **while** not converged and $\min_{u \in \mathcal{V}_T^r} \widehat{h}_T(u) < h$ **do**

        Predict labels of target nodes using $g \circ \phi$.

        Update reliable set $\mathcal{V}_T^r$ by comparing predictions of GNN and MLP.

        Adjust target graph adjacency to obtain $\widetilde{A}^T$.

        Update $\phi$ and $g$ by minimizing the joint objective in Eq. (6).

    **end while**

**Output:** Adjusted adjacency matrices $\widetilde{A}^S$, $\widetilde{A}^T$; trained encoder $\phi$ and classifier $g$.

---

## 5 EXPERIMENTS

**Baselines.** We compare our approach PSAHS against the following representative baselines: feature alignment methods UDA-GCN (Wu et al., 2020), ASN (Zhang et al., 2021a), GraphAlign (Huang et al., 2024), and JHGDA (Shi et al., 2023); structure-shift methods StruRW (Liu et al., 2023) and PairAlign (Liu et al., 2024c); and the homophily-based method HGDA (Fang et al., 2025b).

**Synthetic Experiments.** We evaluate the performance of our method PSAHS under different levels of node homophily shift on the simulated data generated by the stochastic block model (SBM). For each class, we generate an equal number of nodes from three classes. In the source domain, the node attributes are drawn from class-specific 10-dimension Gaussian distributions: the means of the three Gaussians are $[-1, 0, 0_8]$, $[1, 0, 0_8]$ and $[0, 1, 0_8]$ for the source domain, and $[-1.5, 0.5, 0_8]$, $[1.5, -0.5, 0_8]$ and $[0.5, 1.5, 0_8]$ for the target domain, where $0_8$ denotes the 8-dimension all-zero vector. The covariance matrices for the three Gaussians are random rotations of three diagonal matrices: $\texttt{diag}([4_5, (1/4)_5])$, $\texttt{diag}(|\texttt{arange}(10) - 9/2|/(9/2))$, and $\texttt{diag}([4, 1/4, 4, \ldots, 1/4])$, where $\texttt{diag}(\cdot)$ means the diagonal matrix with some vectors. To generate the homophily shift, we fix one domain's node homophily by setting the intra-class probability $p = 0.02$ and the inter-class probability $q = 0.002$, which yields a graph homophily of $0.832$. For the other domain, we iteratively decrease graph homophily by randomly selecting two homophilous edges $(u, u')$ and $(v, v')$, where $Y_u = Y_{u'} \neq Y_v = Y_{v'}$, removing them, and then reconnecting the heterogeneous edges $(u, v)$ and $(u', v')$ to decrease the graph homophily. This procedure is repeated until the graph homophily reaches desired values ranging from $0.8$ to $0.1$. Additional visualizations of attribute distributions are presented in Appendix D.3.2.

Figure 3 presents the GDA accuracy on the synthetic datasets. It reveals that our model PSAHS consistently outperforms baseline methods across varying degrees of node homophily shift, regardless of whether the source or target domain has higher graph homophily. This demonstrates the effectiveness of PSAHS in mitigating node homophily shift.

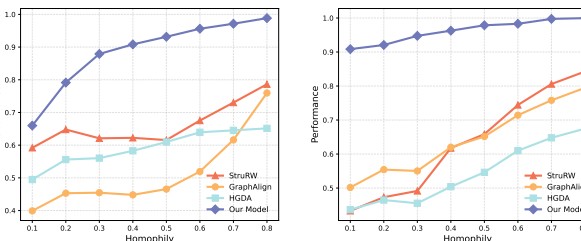

(a) Fixed source with varied target homophily    (b) Fixed target with varied source homophily

Figure 3: Accuracy under different homophily settings.

**Benchmark Datasets.** We conduct comprehensive experiments on four real-world datasets, including `Citation` dataset (Tang et al., 2008; Wu et al., 2022), `Airport` dataset (Ribeiro et al., 2017), `Blog` dataset (Shen et al., 2020a), and `Twitch` dataset (Rozemberczki et al., 2021). The `Citation` dataset consists of two networks, `DBLPv8` (D) and `ACMv9` (A), where nodes correspond to articles and edges represent citation relations. The `Airport` dataset includes three air-traffic networks from the `USA` (U), `Brazil` (B), and `Europe` (E), where each node is an airport, and each edge represents a flight route. The `Twitch` dataset contains six regional gamer networks from `Germany` (DE), `England` (EN), `Spain` (ES), `France` (FR), `Portugal` (PT), and

Russia (RU), with nodes indicating users and edges reflecting friendships. The `Blog` dataset comprises two disjoint social networks, `Blog1` and `Blog2`. Both are derived from `BlogCatalog`, where nodes denote bloggers and edges indicate friendship ties. More details, including the statistics of the datasets, can be checked in Appendix D.3.1.

**Result Analysis.** Tables 2 and 3 show that our method PSAHS consistently outperforms all baselines across 15 GDA tasks, achieving up to a 21.94% improvement on `B2-B1`. These gains highlight the effectiveness of jointly enhancing source homophily and mitigating node homophily shift across domains for GDA on diverse real-world datasets, especially on low-homophily graph datasets such as `Blog` (0.38 average node homophily). In contrast, prior algorithms that rely on the feature aggregation over the *original* graphs perform poorly, as low-homophily structures hinder homophilous feature aggregation during message passing, and mismatched node homophily distributions across domains obstruct knowledge transfer. Details for hyperparameters can be found in Appendix D.4.

Table 2: Performance on `DBLP/ACM` and `Airport` datasets.

| Models | Citation | | Airport | | | | | |
|---|---|---|---|---|---|---|---|---|
| | A-D | D-A | U-E | E-U | B-E | E-B | B-U | U-B |
| UDAGCN | 0.6886 | 0.6391 | 0.4887 | 0.4341 | 0.5077 | 0.4762 | 0.4978 | 0.6122 |
| ASN | 0.7270 | 0.7162 | 0.4645 | 0.4625 | 0.4962 | 0.5903 | 0.4986 | 0.5191 |
| JHGDA | 0.7558 | 0.7322 | 0.5075 | 0.5227 | 0.5664 | 0.7313 | 0.5020 | 0.6927 |
| StruRW | 0.7019 | 0.6657 | 0.5377 | 0.4967 | 0.5606 | 0.6565 | 0.5219 | 0.6284 |
| PairAlign | 0.7524 | 0.7477 | 0.5539 | 0.5428 | 0.5572 | 0.5290 | 0.5278 | 0.6786 |
| GraphAlign | 0.7865 | 0.7506 | 0.5432 | 0.5734 | 0.5880 | 0.7312 | 0.5438 | 0.6290 |
| HGDA | 0.7910 | 0.7560 | 0.5720 | 0.5700 | 0.5840 | 0.7210 | 0.5690 | 0.7210 |
| **PSAHS** | **0.8261** | **0.7583** | **0.5920** | **0.5776** | **0.5948** | **0.7434** | **0.5738** | **0.7245** |

The best and second-best performances are marked as **bold** and underline, respectively.

Table 3: Performance on `Blog` and `Twitch` datasets.

| Models | Blog | | Twitch | | | | |
|---|---|---|---|---|---|---|---|
| | B1-B2 | B2-B1 | DE-EN | DE-ES | DE-FR | DE-PT | DE-RU |
| UDAGCN | 0.4710 | 0.4680 | 0.5397 | 0.5749 | 0.5453 | 0.5532 | 0.6359 |
| ASN | 0.6320 | 0.5240 | 0.5258 | 0.5468 | 0.5279 | 0.5603 | 0.6618 |
| JHGDA | 0.6190 | 0.6430 | 0.5580 | 0.6235 | 0.5921 | 0.6285 | 0.7205 |
| StruRW | 0.6359 | 0.6264 | 0.5481 | 0.6603 | 0.6048 | 0.6396 | 0.7227 |
| PairAlign | 0.6620 | 0.6540 | 0.5669 | 0.6529 | 0.5752 | 0.6250 | 0.7328 |
| HGDA | 0.6830 | 0.6770 | 0.4993 | 0.5443 | 0.5494 | 0.4825 | 0.5460 |
| GraphAlign | 0.4714 | 0.4583 | 0.5602 | 0.6904 | 0.6246 | 0.6574 | 0.7179 |
| **PSAHS** | **0.8805** | **0.8964** | **0.5797** | **0.7129** | **0.6463** | **0.6684** | **0.7413** |

The best and second-best performances are marked as **bold** and underline, respectively.

**Ablation Studies.** We evaluate three variants of our model PSAHS to examine how the choice of domain for structure adjustment affects GDA performance. The variants include DANN (Ganin et al., 2016), a classic adversarial alignment method adapted to GNN encoded representations for GDA; w/o source, which iteratively adjusts the graph structure on the target domain without initial edge adjustment on the source graph; and w/o target, which only refines the source graph structure to reach high homophily while leaving the target graph unchanged.

The ablation results in Table 4 show that both the "w/o source" and "w/o target" variants outperform the baseline DANN, indicating that adjusting either the source or target graph alone can improve GDA performance. More importantly, our full model PSAHS significantly outperforms these single-graph variants, demonstrating the benefits of simultaneously enhancing homophily and mitigating node homophily shift between domains.

Table 4: Ablation study on `Blog` and `Airport` datasets.

| Models | Blog | | Airport | | | | | |
|---|---|---|---|---|---|---|---|---|
| | B1-B2 | B2-B1 | U-E | E-U | B-E | E-B | B-U | U-B |
| DANN | 0.5430 | 0.5625 | 0.4933 | 0.4776 | 0.5099 | 0.6754 | 0.5062 | 0.6547 |
| w/o source | 0.8210 | 0.8288 | 0.5587 | 0.5466 | 0.5558 | 0.6986 | 0.5256 | 0.7075 |
| w/o target | 0.6166 | 0.6017 | 0.5242 | 0.5607 | 0.5434 | 0.7275 | 0.5408 | 0.6918 |
| **PSAHS** | **0.8805** | **0.8964** | **0.5920** | **0.5776** | **0.5948** | **0.7434** | **0.5738** | **0.7245** |

**Model Analysis** Due to the lack of true target labels, pseudo-labels are important for target graph refinement. In this part, we analyze the impact of different pseudo-labeling strategies on GDA

performance. Our model PSAHS adopts consistent label predictions from the GNN and MLP as node pseudo-labels (PLs) and refines edges only for nodes with PLs. For comparison, we consider four variants of PL strategy. GNN_PL directly uses all pseudo labels predicted by the GNN classifier. MLP_PL adopts all pseudo labels predicted by an auxiliary MLP classifier. Curriculum_PL adopts a progressive scheme. Specifically, it begins with adjusting the edges for the top $20\%$ most confident target nodes for graph refinement and gradually increases the ratio to $80\%$ as training proceeds. Prototype_PL employs prototypical denoising, where pseudo-labels are reweighted based on their distances to class prototypes that are updated online via moving averages.

Figure 4 reports the GDA performance on `Blog` dataset. Our model PSAHS outperforms all variants, demonstrating the benefits of integrating both structural and attribute-based views. Since structure adjustment is applied only to the low-homophily nodes that are vulnerable to the disruptive effect of heterophilic edges, leveraging the auxiliary MLP view, which relies solely on attribute information, yields more accurate label predictions for these nodes and drives a clear performance gain.

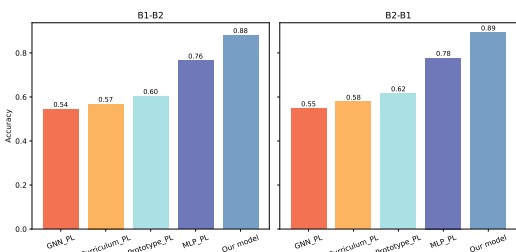

Figure 4: GDA performance of different PL strategies.

## 6 CONCLUSION AND FUTURE WORK

In this paper, we investigated the challenge of *node homophily shift* in GDA, a structural mismatch that hinders cross-domain transfer even when feature distributions are aligned. We proposed a progressive structure adjustment framework that alternates between source-side homophily enhancement, target-side homophily alignment guided by pseudo-labels, and cross-domain representation alignment via adversarial training. Our theoretical analysis established an explicit connection between homophily distributions and the target error bound, thereby motivating structural refinement as a principled approach. Extensive experiments on both synthetic and real-world benchmarks demonstrated that the proposed method consistently outperforms strong baselines, with particularly large improvements under severe homophily mismatch. These results highlight the critical role of structural alignment in enabling effective cross-graph transfer.

While our framework effectively reduces node homophily shift across domains, addressing fairness and subgroup generalization under node homophily shift across domains remains a promising direction for future work. Such investigations could offer deeper insights into the equitable deployment of GDA methods in real-world applications.

## ETHICS STATEMENT

This work makes use of publicly available datasets and models. No private or sensitive data is involved, and no harmful content is included. Therefore, we believe this paper does not raise any ethical concerns.

## REPRODUCIBILITY STATEMENT

Implementation details for our proposed algorithm are provided in Appendix D.1, and the corresponding code is available via the anonymous link `https://anonymous.4open.science/r/PSAHS`. Descriptions and statistics of all datasets are presented in Section 5 and Appendix D.3, with the data processing scripts also provided via the anonymous link. Full proofs of the theoretical claims are included in Appendix C.

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

This appendix complements the main text by providing additional theoretical analysis, detailed proofs, and extended experimental results. Specifically, Appendix A introduces the related works about graph domain adaptation and graph learning with homophily. Appendix B analyzes how node homophily ratios influence class separability under the CSBM-structure model. Appendix C contains the formal proofs of the theorems presented in Sections 3 and 4. Finally, Appendix D reports supplementary experimental results, including dataset descriptions, implementation details, model analyses, and parameter sensitivity studies.

# A RELATED WORKS

## A.1 GRAPH DOMAIN ADAPTATION

Early research on graph domain adaptation (GDA), often referred to as cross-network classification, mainly focused on learning shared features across networks based solely on graph structures (Shen et al., 2020b). With the development of graph neural networks (GNNs), GDA research has expanded to attributed graphs, where both structural and attribute shifts are taken into consideration (Liu et al., 2024b). These methods typically integrate GNNs with traditional domain adaptation strategies to learn transferable node representations. For instance, adversarial learning has been employed to extract domain-invariant features (Zhang et al., 2019; Wu et al., 2020; Zhang et al., 2021b). Other approaches adopt direct feature alignment techniques that use various distance metrics to encourage feature consistency, such as maximum mean discrepancy (MMD) (Shi et al., 2023), total variation distance (Chen et al., 2025), graph subtree discrepancy (Wu et al., 2023), and optimal transport distance (Chen et al., 2020). However, the standard GNN architectures are often inadequate for capturing the complex structures inherent in graph data. To address this limitation, some methods enhance node features by incorporating richer structural information, including high-order structures (Dan et al., 2024; Yin et al., 2025), substructures (Luo et al., 2024), and spectral properties (You et al., 2023; Xiao et al., 2024). Meanwhile, the message-passing mechanism of GNNs can also be improved to better support GDA (Liu et al., 2024a).

Most GDA methods in the feature space typically borrow conventional alignment strategies from other domains and often overlook the unique properties of graph data. Recently, increasing efforts have been made to directly tackle structural shifts and develop adaptation methods on the input graph structures. For example, Huang et al. (2024) adopts a data-centric approach that constructs a smaller yet more transferable source graph to better align with the target graph. Several studies address conditional structure shifts induced by labels and propose reweighting strategies to adjust graph edges accordingly (Liu et al., 2023; 2024c). Fang et al. (2025a) further considers attribute shifts and combines topology and attribute graphs for GDA, while Fang et al. (2025b) emphasizes the influence of graph homophily and develops mixed graph filters to improve adaptation.

## A.2 GRAPH LEARNING WITH HOMOPHILY AND HETEROPHILY

Developing GNNs in heterophilic graphs has received increasing attention. The primary goal is to investigate the consistency of raw graph structure and node label similarities, where homophily originally refers to the matching of edges with label similarities and vice versa (Zhu et al., 2020). The definition of homophily varies in different settings, such as local-global homophily (Li et al., 2022), and structural-feature-label homophily (Zheng et al., 2024).

In general, current research for homophily GNNs can be categorized as data-based and model-based methods. Data-based methods focus on improving the homophily ratios by refining existing or discovering new neighbors for a given node. The intuitive strategy is to incorporate higher-order neighbors with the same labels (Li et al., 2022). Particularly, Zhu et al. (2020) has theoretically demonstrated that the 2-hop neighbors of nodes are homophily-dominant and can therefore facilitate the feature aggregation in GNNs. Zheng et al. (2023) constructs a complementary graph to discover potential neighbors and uses the complemented graph convolution to leverage both homophily and heterophily connections. The graphs can also be rewired or reconstructed to high-homophily counterparts by further calculating feature distances (Li et al., 2023) or structural similarities (Suresh et al., 2021).

The model-based methods aim to develop new aggregation and updating processes in GNNs to strengthen homophilic information and debilitate heterophilic information. A line of methods ex-

tends the uniform message passing schemes into diverse ones (Yang et al., 2021; Chanpuriya & Musco, 2022), such as combining low-pass filter in GNNs with high-pass filters (Luan et al., 2022; Duan et al., 2024) and heat kernels (Li et al., 2024), incorporating homophily-enhanced neighbor aggregation (Wang et al., 2022; Jin et al., 2022). The architecture of GNN can also be adjusted to fit the homophilic and heterophilic patterns in graphs. For example, Yan et al. (2023) redefines the number of aggregation layers in GNNs as a tunable real number and shows that adaptive layer depth can better filter low/high signals in homophilic/heterophilic graphs.

## B   EFFECT OF NODE HOMOPHILY ON CLASS SEPARABILITY

In this section, we examine how node homophily ratios influence the separability of aggregated features under the contextual stochastic block model with structure (CSBM-S) proposed by Mao et al. (2023).

Specifically, we generate two disjoint node sets, $\mathcal{C}_1$ and $\mathcal{C}_2$. Each node attribute $X_u$ is sampled from $\mathcal{N}(\mu_i, I)$ with $i \in \{1, 2\}$, and the class prior is balanced, i.e., $P(Y = 1) = P(Y = 2) = 1/2$. To induce different distributions of node homophily ratios, each set $\mathcal{C}_i$ is divided into two groups:

- $\mathcal{C}_i^1$: high-homophily nodes with intra-class and inter-class edge probabilities $p_1 > q_1$;
- $\mathcal{C}_i^2$: low-homophily nodes with probabilities $p_2 < q_2$.

We further assume that all nodes follow the same degree distribution, ensuring $p_1 + q_1 = p_2 + q_2$.

**Aggregated Feature Distributions.**   Let $F = D^{-1}AX$ denote the aggregated node features. The means of these features for the two homophily groups can be expressed as

$$f_i^j \sim \mathcal{N}\left(\frac{p_j\mu_1 + q_j\mu_2}{p_j + q_j}, \frac{I}{d_i}\right), \quad i \in \mathcal{C}_1^j; \qquad f_i^j \sim \mathcal{N}\left(\frac{q_j\mu_1 + p_j\mu_2}{p_j + q_j}, \frac{I}{d_i}\right), \quad i \in \mathcal{C}_2^j, \quad (7)$$

where $f_i^j$ denotes the aggregated features of group $j$ in class $i$.

From Eq. (7), when $q_1 = p_2$ and $p_1 = q_2$, we obtain $\mathbb{E}f_i^1 = \mathbb{E}f_{i'}^2$ and $\mathbb{E}f_i^2 = \mathbb{E}f_{i'}^1$, where $i \in \mathcal{C}_1$, $i' \in \mathcal{C}_2$. This implies that the aggregated features of high-homophily nodes in one class overlap with those of low-homophily nodes in the other class, making the two classes indistinguishable after aggregation.

**Separability Analysis.**   To investigate which graph structures in the CSBM-S model lead to good class separability, we analyze the feature margin. As shown in Theorem B.1, the margin between two classes, defined as

$$\mathcal{M} := \min_{j, j' \in \{1, 2\}} \left\|\mathbb{E}f_i^j - \mathbb{E}f_{i'}^{j'}\right\|_2, \qquad i \in \mathcal{C}_1^j, \; i' \in \mathcal{C}_2^{j'}, \quad (8)$$

is maximized when all nodes are highly homophilous.

**Theorem B.1** (Largest Margin when All Nodes are High-homophily). *Consider the CSBM-S model described above and let the class margin be defined as in Eq. (8). Then*

$$\max_{\mathcal{G}} \mathcal{M} = \|\mu_1 - \mu_2\|_2,$$

*and the maximum is attained if and only if*

$$q_1 = 0, \quad \mathcal{C}_1^2 = \mathcal{C}_2^2 = \emptyset. \quad (9)$$

*That is, the graph exhibits high-homophily across all nodes, with no low-homophily groups present.*

*Proof of Theorem B.1.* From Eq. (7), the mean of the aggregated features for a node in group $j$ of class 1 is

$$\mathbb{E}f_{(1)}^j = \frac{p_j}{p_j + q_j}\mu_1 + \frac{q_j}{p_j + q_j}\mu_2,$$

and for a node in group $j$ of class 2,

$$\mathbb{E}f_{(2)}^j = \frac{q_j}{p_j + q_j}\mu_1 + \frac{p_j}{p_j + q_j}\mu_2.$$

Thus both expectations lie on the line segment between $\mu_1$ and $\mu_2$, i.e.,

$$\mathbb{E}f \in \mathrm{conv}\{\mu_1, \mu_2\}.$$

Define $\alpha_j := \frac{p_j}{p_j + q_j}$. Then the expectations can be written as

$$\mathbb{E}f_{(1)}^j = \alpha_j\mu_1 + (1 - \alpha_j)\mu_2, \qquad \mathbb{E}f_{(2)}^j = (1 - \alpha_j)\mu_1 + \alpha_j\mu_2.$$

The distance between the aggregated means of group $j$ in class 1 and group $j'$ in class 2 is

$$\left\|\mathbb{E}f_{(1)}^j - \mathbb{E}f_{(2)}^{j'}\right\| = \left|\alpha_j + \alpha_{j'} - 1\right|\|\mu_1 - \mu_2\|.$$

By definition,

$$\mathcal{M} = \min_{j,j'}\left\|\mathbb{E}f_{(1)}^j - \mathbb{E}f_{(2)}^{j'}\right\| \le \|\mu_1 - \mu_2\|,$$

with equality if and only if $\alpha_j = 1$ for every nonempty group $j$. Equivalently, this requires $q_j = 0$ for all such groups. Under the degree constraint $q_2 > p_2$, the condition $q_j = 0$ forces the low-homophily groups $\mathcal{C}_1^2, \mathcal{C}_2^2$ to be empty.

Therefore, under high-homophily we obtain

$$\mathcal{M} = \|\mu_1 - \mu_2\|_2,$$

which is the largest possible margin. This completes the proof. $\qquad\square$

Theorem B.1 shows that maintaining consistently high homophily ratios across all nodes—rather than mixing high- and low-homophily nodes—maximizes class separability. Moreover, in a highly homophilous graph, increasing the degree $d_i$ of node $i$ reduces the variance of $f_i^j$, thereby further lowering the classification error. This leads to Theorem B.2.

**Theorem B.2.** *Consider a linear classifier $h(x) = \mathrm{sign}(w^\top x + b)$ trained on aggregated features $f_i^j$. Let $\epsilon(h)$ denote its misclassification error. Then*

$$\min_{\mathcal{G}} \epsilon(h) \quad \text{is attained when} \quad (u, v) \in E \iff Y_u = Y_v.$$

*Proof of Theorem B.2.* From Theorem B.1, the largest class margin $\|\mu_1 - \mu_2\|_2$ is obtained under Eq. (9). The linear classifier for two Gaussian $\mathcal{N}(\mu_k, \sigma_k^2 I)$, $k = 1, 2$ is given by

$$h(x) := \mathrm{sign}\left(\frac{\mu_1 - \mu_2}{\|\mu_1 - \mu_2\|}x - t\right),$$

where $t \in \mathbb{R}$ is the bias parameter of the optimal classifier. The risk of the classifier is

$$\epsilon(h) = \frac{1}{2}\Phi\left(\frac{t - \|\mu_1 - \mu_2\|}{\sigma_1}\right) + \frac{1}{2}\left(1 - \Phi\left(\frac{t}{\sigma_2}\right)\right),$$

where $\Phi$ is the standard Gaussian cumulative density function. From Eq. (7), for any $i \in C_1$ and $i' \in C_2$, we have

$$\sigma_1^2 = 1/d_1, \qquad \sigma_2^2 = 1/d_{i'}.$$

Therefore, the minimal risk $\epsilon(h)$ decreases as the degree $d_i$ increases for all nodes. Hence minimizing $\epsilon(h)$ requires:

 *(i)* maximizing the margin $\mathcal{M}$ (achieved when all nodes are high-homophily), and

 *(ii)* maximizing node degrees $d_i$ while preserving homophily.

This is achieved when each node is connected to all nodes with the same label, which proves the assertion. $\square$

Theorem B.2 highlights that, beyond removing inter-class edges, class separability also benefits from adding more intra-class edges. This directly motivates the source-domain adjustment strategy in Section 4.1, where inter-class edges are pruned and additional intra-class edges are introduced for low-homophily nodes. Since source labels are fully observable, such modifications are both natural and straightforward to implement.

## C PROOFS

### C.1 PROOFS RELATED TO SECTION 3

The proof of Theorem 3.1 builds on the PAC-Bayesian framework for domain adaptation. We start from the generalization bound in Mao et al. (2023), which relates the target error to the source error, a KL-divergence term, and a discrepancy term between source and target. By incorporating the adjusted homophily ratios and node representations, and applying the inequality in Fang et al. (2025b), the discrepancy can be bounded by the average difference in homophily and representation across domains. Choosing $\lambda = n_s^{2\alpha}$ and following the concentration arguments of Mao et al. (2023), we obtain the final bound where the target risk is controlled by: (i) the empirical source margin loss, (ii) the cross-domain homophily difference, and (iii) the feature representation discrepancy. This shows that structural refinement (to adjust homophily) and representation alignment (to reduce embedding discrepancy) are both crucial for tightening the bound.

**Assumption C.1** (Data follows Generalized CSBM-S model assumption). *The graph data is generated from the Generalized CSBM-S model.*

**Definition C.2** (Distance To Training Set and Near Set). Define the distance from the target graph to the source graph as
$$\epsilon := \max_{j \in V_t} \min_{i \in V_s} \|f_i(X, A) - f_j(X, A)\|_2.$$

Further, for each $i \in V_s$, define the near set of $i$ with respect to $V_t$ as
$$V_t^{(i)} := \{j \in V_t \mid \|f_i(X, A) - f_j(X, A)\|_2 \le \epsilon\}.$$

Clearly,
$$V_t = \cup_{i \in V_s} V_t^{(i)}.$$

**Assumption C.3** (Equal-Sized and Disjoint Near Sets ). *Assume the near sets of each $i \in V_s$ with respect to $V_t$ are disjoint and have the same size $s \in \mathbb{N}^+$.*

Assumption C.3 assumes that the target nodes can be divided into equally sized partitions, where all nodes in each partition share a same closest source node. It assumes that target nodes are closely aligned with the respective source node, while distant to the other source nodes.

**Assumption C.4** (Concentrated Expected Loss Difference). *Let $P$ be a distribution on $\mathcal{H}$, defined by sampling the vectorized MLP parameters from $\mathcal{N}(0, \sigma^2 I)$ for some $\sigma^2 \le \frac{(\gamma/8\epsilon_m)^{2/L}}{2b(\lambda N_0^{-\alpha} + \ln 2bL)}$. For any $L$-layer GNN classifier $h \in \mathcal{H}$ with model parameters $W_1^h, \ldots, W_L^h$, define $T_h := \max_{l=1,\ldots,L} \|W_l^h\|_2$. Assume that there exists some $0 < \alpha < \frac{1}{4}$ satisfying*

$$\Pr_{h \sim P} \left( \mathcal{L}_m^{\gamma/4}(h) - \mathcal{L}_0^{\gamma/2}(h) > N_0^{-\alpha} + cK\epsilon_m \mid T_h^L \epsilon_m > \frac{\gamma}{8} \right) \le e^{-N_0^{2\alpha}}.$$

Assumption C.4 postulates that the expected margin loss on the target graph, is not significantly larger that on the train node subgroup, as the number of source graph becomes larger.

**Theorem** (Restate of Theorem 3.1). *Under the SGN model and assumptions C.1-C.4, let $g \circ \phi$ be a classifier from the hypothesis space of GNN classifiers. Denote by $\widetilde{f}_u$ (resp. $\widetilde{f}_v$) the aggregated feature of a source node $u \in \mathcal{V}_S$ (resp. target node $v \in \mathcal{V}_T$) under the adjusted graph structure $\widetilde{A}^S$ (resp. $\widetilde{A}^T$). Similarly, let $\widetilde{h}_S(u)$ and $\widetilde{h}_T(v)$ denote the corresponding node homophily ratios. Then for any $\gamma > 0$, $\delta \in (0, 1)$, $\alpha \in (0, 1/4)$, and sufficiently large $n_s$, there exists a constant $c$*

*independent of $n$ such that with probability at least $1 - \delta$, the target margin loss $\mathcal{R}_T(g \circ \phi)$, can be upper bounded by*

$$\widehat{\mathcal{R}}_S^{\gamma}(g \circ \phi) + c \left( \frac{1}{n_s n_t} \sum_{u \in \mathcal{V}_S} \sum_{v \in \mathcal{V}_T} \left( |\widetilde{h}_S(u) - \widetilde{h}_T(v)| + \|\widetilde{f}_u - \widetilde{f}_v\|_2 \right) + \frac{1}{n_s^{\alpha}} + \frac{\ln(1/\delta)}{n_s^{2\alpha}} \right). \quad (10)$$

*Proof of Theorem 3.1.* Combining Lemma 6 and Theorem 2 in Mao et al. (2023), we obtain that

$$\mathcal{R}_T(g \circ \phi) - \widehat{\mathcal{R}}_S^{\gamma}(g \circ \phi) \leq \frac{1}{\lambda} \left( 2 \big( D_{KL}(Q \| P) + 1 \big) + \ln \left( \frac{1}{\delta} \right) + \frac{\lambda^2}{4 n_s} + D_{T,S}^{\gamma/2}(P, \lambda) \right) \quad (11)$$

holds with probability at least $1 - \delta$, where

$$D_{T,S}^{\gamma/2}(P, \lambda) := \ln \mathbb{E}_{\phi \sim P} \exp \left( \lambda \big( \mathcal{R}_T^{\gamma/4}(g \circ \phi) - \mathcal{R}_S^{\gamma/2}(g \circ \phi) \big) \right).$$

By applying Inequality (22) in Fang et al. (2025b) to the adjusted node homophily $\widetilde{h}_u$ and the adjusted GNN representation $\widetilde{f}_u$, we have

$$D_{T,S}^{\gamma/2}(P, \lambda) \leq \frac{1}{n_s n_t} \sum_{u \in \mathcal{V}_S} \sum_{v \in \mathcal{V}_T} \left( \ln 3 + \frac{2\lambda\rho}{\sqrt{2\pi}\sigma} \big( \|\widetilde{f}_u - \widetilde{f}_v\|_2 + \rho \cdot |\widetilde{h}_u - \widetilde{h}_v| \big) \right), \quad (12)$$

where $\rho := \|\mu_1 - \mu_2\|_2$.

Substituting Eq. (12) into Eq. (11), we obtain

$$\mathcal{R}_T(\phi) - \widehat{\mathcal{R}}_S^{\gamma}(\phi) \leq \frac{1}{\lambda} \Bigg( 2(D_{KL}(Q\|P) + 1) + \ln \left( \frac{1}{\delta} \right) + \frac{\lambda^2}{4 n_s}$$

$$+ \frac{1}{n_s n_t} \sum_{u \in \mathcal{V}_S} \sum_{v \in \mathcal{V}_T} \left( \ln 3 + \frac{2\lambda\rho}{\sqrt{2\pi}\sigma} \big( \|\widetilde{f}_u - \widetilde{f}_v\|_2 + \rho \cdot |\widetilde{h}_u - \widetilde{h}_v| \big) \right) \Bigg).$$

Next, set $\lambda = n_s^{2\alpha}$ and apply the same analysis as in Inequality (47) of Mao et al. (2023). This yields

$$\mathcal{R}_T(\phi) - \widehat{\mathcal{R}}_S^{\gamma}(\phi)$$

$$\leq c' \left( \frac{\rho}{n_s n_t} \sum_{u \in \mathcal{V}_S} \sum_{v \in \mathcal{V}_T} \big( \|\widetilde{f}_u - \widetilde{f}_v\|_2 + \rho \cdot |\widetilde{h}_u - \widetilde{h}_v| \big) + \frac{\sum_{\ell=1}^{L} \|W_\ell\|_F^2}{n_s^{\alpha}} + \frac{1}{n_s^{1-2\alpha}} + \frac{\ln(1/\delta)}{n_s^{2\alpha}} \right)$$

$$\leq c \left( \frac{1}{n_s n_t} \sum_{u \in \mathcal{V}_S} \sum_{v \in \mathcal{V}_T} \big( \|\widetilde{f}_u - \widetilde{f}_v\|_2 + \rho \cdot |\widetilde{h}_u - \widetilde{h}_v| \big) + \frac{1}{n_s^{\alpha}} + \frac{\ln(1/\delta)}{n_s^{2\alpha}} \right),$$

with probability at least $1 - \delta$, where $c$ and $c'$ are constants depending on $\gamma$, the maximum norm of node representations, and the maximum hidden-layer width. This finishes the proof. $\square$

## C.2 PROOFS RELATED TO SECTION 4

*Proof of Theorem 4.1.* By the definition of $\widetilde{h}_S(u)$, we have

$$\widetilde{h}_S(u) = \frac{\alpha(1 - h_S(u)) + (1 + \alpha)h_S(u)}{\alpha(1 - h_S(u)) + (1 + \alpha)h_S(u) + (1 - \alpha)(1 - h_S(u))} = \frac{\alpha + h_S(u)}{1 + \alpha h_S(u)}.$$

If $\alpha > \frac{h - h_S(u)}{1 - h \cdot h_S(u)}$, then

$$\frac{\alpha + h_S(u)}{1 + \alpha h_S(u)} > h,$$

which implies $\widetilde{h}_S(u) > h$. $\square$

*Proof of Theorem 4.2.* By Theorem 4.1, for any source node $u$ and target node $v$ with $h_S(u) < h$ and $h_T(v) < h$, we have $\widetilde{h}_S(u) = h$ and $\widetilde{h}_T(v) = h$. Hence,

$$|\widetilde{h}_S(u) - \widetilde{h}_T(v)| = 0 < |h_S(u) - h_T(v)|.$$

For $h_S(u) \geq h$ and $h_T(v) < h$, we obtain $\widetilde{h}_S(u) = h_S(u)$ and $\widetilde{h}_T(v) = h$, which yields

$$|\widetilde{h}_S(u) - \widetilde{h}_T(v)| = h_S(u) - h \leq h_S(u) - h_T(v) = |h_S(u) - h_T(v)|.$$

Moreover, $|\widetilde{h}_S(u) - \widetilde{h}_T(v)|$ is non-decreasing in $h$.

For $h_S(u) < h$ and $h_T(v) \geq h$, we get $\widetilde{h}_S(u) = h$ and $\widetilde{h}_T(v) = h_T(v)$, so

$$|\widetilde{h}_S(u) - \widetilde{h}_T(v)| = h_T(v) - h \leq h_T(v) - h_S(u) = |h_S(u) - h_T(v)|.$$

Finally, when $h_S(u) \geq h$ and $h_T(v) \geq h$, we have $\widetilde{h}_S(u) = h_S(u)$ and $\widetilde{h}_T(v) = h_T(v)$, giving

$$|\widetilde{h}_S(u) - \widetilde{h}_T(v)| = |h_S(u) - h_T(v)|.$$

These cases together establish the claim. $\qquad\square$

# D COMPLEMENTARY EXPERIMENTS

## D.1 EXPERIMENTAL SETUP

The experiments are implemented using the `PyTorch` platform on a workstation equipped with an `Intel(R) Core(TM) i7-14700K CPU@3.40GHz` and a `NVIDIA GeForce RTX 4080 16GB GPU`. For all datasets, we adopt a $k$-layer GNN, where $k$ ranges from 2 to 5 and the hidden dimension is selected from $32, 64, 128, 512$. Both the domain discriminator and the classifier are implemented as two-layer MLPs with hidden dimensions chosen from $16, 32, 64, 128$. To improve the quality of pseudo labels on the target dataset, we pretrain an auxiliary MLP classifier with a 128-64-64 architecture on the source domain, which provides pseudo labels from a complementary perspective. We select the learning rate in $\{0.0001, 0.001, 0.003, 0.01\}$. The setting of hyperparameters $\gamma_{\text{RA}}$ follows the schedule: $\min\{2/(1 + e^{-10p}) - 1, 0.1\}$, where $p$ changes from 0 to 1 during the training process, as described in Ganin et al. (2016). Additionally, the parameter grid for the homophily threshold $h$ is $\{0.5, 0.6, \ldots, 0.9, 1.0\}$. Guided by Theorem 4.2, for source nodes $u$ with $h_S(u) < h$, we take $\alpha_u = (h - h_S(u))/(1 - h \cdot h_S(u))$ and for target nodes $v$ with $\widehat{h}_T(v) < h$ where $\widehat{h}_T$ is defined as in Eq. (4), we take $\alpha_v = (h - \widehat{h}_T(v))/(1 - h \cdot \widehat{h}_T(v))$.

The total number of training epochs is set to 300. Since the pseudo-labels generated for the target domain are unreliable at the early stage, we adopt a warm-up strategy to stabilize training. Specifically, we introduce a starting epoch $e$ and a reweighting frequency $t$, so that the model can gradually adapt to the evolving graph structure instead of being continuously updated from the beginning. The starting epoch determines when to begin imposing edge weights on the target graph. The reweighting frequency specifies how often the edge weights are updated. The search spaces for $e$ and $t$ are $\{100, 150, 200, 250\}$ and $\{1, 5, 10, 15\}$. We repeatedly train and test our model for five times with the same partition of dataset and then report the average of accuracy.

## D.2 COMPARED METHODS

We compare our method with the following representative methods. UDA-GCN (Wu et al., 2020) develops a dual graph convolutional network that jointly exploits local and global consistency for better adaptation. ASN (Zhang et al., 2021a) improves node representations by disentangling domain-specific and domain-invariant factors through private and shared encoders. JHGDA (Shi et al., 2023) explores information from different levels of network hierarchy by hierarchical pooling model. StruRW (Liu et al., 2023) and PairAlign (Liu et al., 2024c) reweight edges in the source graph to reduce the conditional shift of neighborhoods. GraphAlign (Huang et al., 2024) generates a small yet transferable graph that aligns with the target via MMD and preserves transferable knowledge through gradient matching. HGDA (Fang et al., 2025b) mitigates the homophily shift by aligning multi-view feature representations across domains.

Table 5: Dataset Statistics.

| Dataset | # Domains | # Nodes | # Edges | # Node_Homo | # Edge_Homo | # Feat Dims | # Labels |
|---|---|---|---|---|---|---|---|
| Airport | USA | 1,190 | 27,198 | 0.3728 | 0.6978 | 241 | 4 |
| | BRAZIL | 131 | 2,148 | 0.2478 | 0.4683 | | |
| | EUROPE | 399 | 11,990 | 0.2195 | 0.4048 | | |
| Blog | Blog1 | 2,300 | 66,942 | 0.3887 | 0.3991 | 8,189 | 6 |
| | Blog2 | 2,896 | 107,672 | 0.3728 | 0.4002 | | |
| Citation | DBLPv8 | 5,578 | 7,341 | 0.9750 | 0.9654 | 7,537 | 6 |
| | ACMv9 | 7,410 | 11,135 | 0.8179 | 0.8335 | | |
| Twitch | England | 7,126 | 35,324 | 0.5536 | 0.5560 | 3,170 | 2 |
| | Germany | 9,498 | 153,138 | 0.5974 | 0.6322 | | |
| | France | 6,566 | 65,955 | 0.5716 | 0.5595 | | |
| | Russia | 4,385 | 37,304 | 0.6300 | 0.6176 | | |
| | Spain | 4,648 | 59,382 | 0.6137 | 0.5800 | | |
| | Portugal | 1,912 | 31,299 | 0.5945 | 0.5708 | | |

### D.3  DATASET

#### D.3.1  DATASET STATISTICS

Table 5 presents the basic statistics of each dataset, such as the numbers of nodes, the numbers of edges, feature dimensions, and labels. In addition, we report the average node and edge homophily, providing a measure of dataset homophily.

#### D.3.2  VISUALIZATION FOR SYNTHETIC DATASET

We take a two-dimensional example with the same attribute generation procedure as in the main text. For each class, we generate an equal number of nodes from three classes. In the source domain, the node attributes are drawn from class-specific 2-dimension Gaussian distributions: the means of the three Gaussians are $[-1, 0]$, $[1, 0]$ and $[0, 1]$ for the source domain, and $[-1.5, 0.5]$, $[1.5, -0.5]$ and $[0.5, 1.5]$ for the target domain. The covariance matrices for the three Gaussians are random rotations of three diagonal matrices: $\texttt{diag}([4_5, (1/4)_5])$, $\texttt{diag}(|\texttt{arange}(10) - 9/2|/(9/2))$, and $\texttt{diag}([4, 1/4, 4, \ldots, 1/4])$, where $\texttt{diag}(\cdot)$ means the diagonal matrix with some vectors.

Figure 5(a) presents class-wise attribute contours on source (dashed) and target (solid) domains. For each class, we estimate a Gaussian density from its samples and plot two equal-probability contour levels. Differences in separation, overlap, and orientation across domains reflect both mean and covariance shifts in the conditional distributions $P(X \mid Y)$. To further visualize marginal distribution shift, Figure 5(b) (right) applies kernel density estimation (KDE) on all node attributes regardless of labels. The resulting contours approximate the overall $P_X$ in the source and target domains, providing a more realistic representation of domain-level attribute distribution than Gaussian formulations.

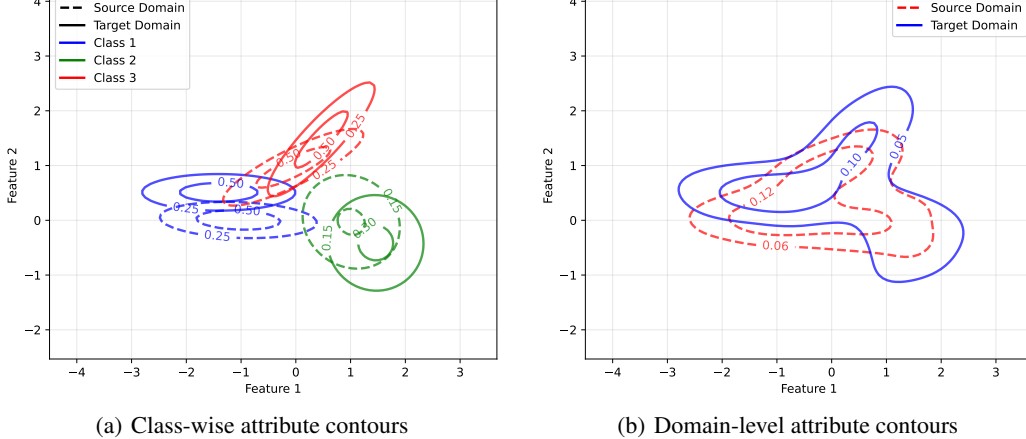

(a) Class-wise attribute contours          (b) Domain-level attribute contours

Figure 5: Visualization of synthetic data.

### D.4 PARAMETER ANALYSIS

In this part, we analyze the influence of the homophily threshold $h$ on the performance of our model. As shown in Figure 6, simply increasing the homophily threshold $h$ does not always guarantee better performance. On datasets such as `Blog`, adding more homophilous edges steadily improves performance. In contrast, on datasets like `Airport`, performance peaks at an intermediate threshold and then declines. We attribute this to the amplification of noise in pseudo-labels. When the target domain contains higher label uncertainty, enforcing too many additional homophilous edges propagates and magnifies such errors, which compromises the benefits of structural refinement.

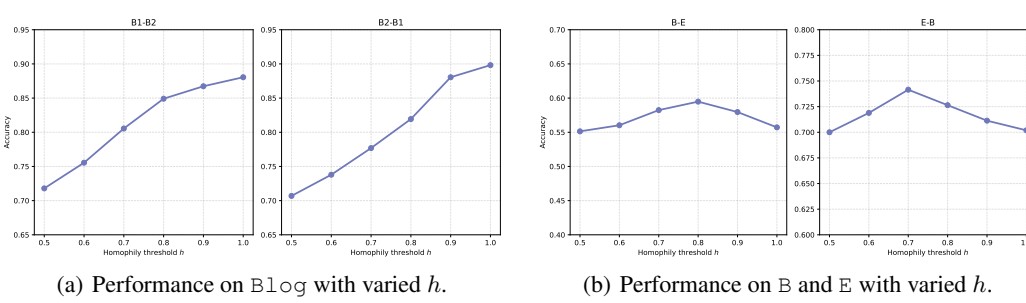

(a) Performance on `Blog` with varied $h$.    (b) Performance on `B` and `E` with varied $h$.

Figure 6: Parameter Analysis of homophily threshold $h$.

## THE USE OF LARGE LANGUAGE MODELS (LLM)

We commit to using LLMs for text polishing based on prompts. All polished text are double-checked by authors to ensure accuracy, avoid over-claims, and prevent confusion.

