# OpenReview forum: "Progressive Graph Structure Adjustment for Homophily Shift in Graph Domain Adaptation"
_ICLR.cc/2026/Conference — Submitted to ICLR 2026_

### Official Review · Reviewer_fChT · 2025-10-27

**Soundness:** 2
**Presentation:** 2
**Contribution:** 2
**Rating:** 4
**Confidence:** 5

**Summary:**

The paper addresses node homophily shift in Graph Domain Adaptation (GDA), where the tendency of nodes to connect with same-label neighbors differs between source and target graphs. The authors propose PSAHS (Progressive Structure Adjustment for Homophily Shift), which iteratively adjusts graph structures to enhance source homophily and align target homophily distributions through pseudo-labels, combined with domain-adversarial representation alignment. The method is theoretically motivated by an error bound linking target performance to homophily shift.

**Strengths:**

* **Strong theoretical foundation**: The derivation of Theorem 3.1 establishes a clear connection between homophily shift and target error bounds, providing principled motivation for the proposed structural adjustment approach.

* **Comprehensive methodology**: The three-stage framework elegantly combines source homophily enhancement, target structure refinement using consistent GNN-MLP predictions, and representation alignment in a mutually reinforcing manner.

* **Consistent empirical improvements**: The method demonstrates substantial gains across diverse benchmarks (up to 21.94% on Blog dataset), with particularly strong performance under severe homophily mismatch scenarios.

**Weaknesses:**

* **Limited technical novelty in alignment component**: The domain-adversarial alignment loss follows standard DANN formulation without clear justification for why this specific choice is optimal. How does this compare theoretically and empirically to alternatives like MMD-based alignment used in GraphAlign? What are the trade-offs?

* **Insufficient analysis of baseline behavior**: Figure 2 shows intriguing differences in how StruRW, GraphAlign, and HGDA respond to fixed source vs. fixed target homophily scenarios, but the paper lacks explanation. Why does GraphAlign maintain relatively stable performance while HGDA shows dramatic degradation at certain homophily levels? Understanding these patterns would strengthen the contribution.

* **Computational complexity not addressed**: The iterative refinement process with repeated pseudo-label generation and structure adjustment likely incurs significant overhead compared to baselines, yet runtime analysis is absent.

* **Limited ablation on key design choices**: The choice of using GNN-MLP agreement for reliable node selection lacks thorough justification. How sensitive is performance to this threshold? What percentage of nodes typically qualify as "reliable"?

I would consider raising my score if the authors adequately address these concerns.

**Questions:**

See Weaknesses

---

> ### Author Response · Authors · 2025-11-28
>
> **W1.** Limited technical novelty in alignment component: The domain-adversarial alignment loss follows standard DANN formulation without clear justification for why this specific choice is optimal. How does this compare theoretically and empirically to alternatives like MMD-based alignment used in GraphAlign? What are the trade-offs?
>
> **AW1.**
> We clarify that the novelty of our work does **not** lie in the choice of a specific feature alignment tool (e.g., DANN). Instead, the main contributions of PSAHS concern (i) a novel insight that homophily shift can be directly corrected by adjusting graph structures, rather than merely aligning features, (ii) a theory-driven formulation that dictates *how* graph structures should be adjusted, and (iii) a progressive, homophily-aware graph adjustment algorithm that has not appeared in prior GDA research. Below, we elaborate on these points and also discuss why DANN is chosen as an instantiation of the general feature-aligner component.
>
> **(1) Conceptual Novelty: Directly Revising Homophily via Graph Structure Adjustment**
>
> Prior work addressing homophily shift, like GraphAlign [1], implicitly treats node homophily as an intrinsic property of the graph. As a result, these methods attempt to mitigate homophily shift *indirectly*, typically via multi-view representation alignment (e.g., aligning attribute-view and topology-view representations). However, they do **not** attempt to modify the homophily distributions themselves.
>
> Our key insight is that **node homophily is not fixed**: it depends jointly on node labels and graph structure.
> Since revising labels is risky—especially in the label-scarce target domain—**graph structure is the only reliable and theoretically justified lever** for directly modifying homophily.
>
> Thus, PSAHS is the first GDA framework that:
>
> * explicitly treats homophily shift as a *structural phenomenon*
> * directly *revises node-level homophily ratios* in both domains
> * uses this understanding to design a principled graph-adjustment strategy
>
> This differs fundamentally from the feature alignment philosophy underlying GraphAlign and other GDA baselines.
>
> **(2) Framework Novelty: Theoretical Motivation for Graph Structure Adjustment**
>
> Our method is guided by the target-domain generalization bound in **Theorem 3.1**, which decomposes the target error of a GNN classifier trained on adjusted graphs into:
>
> 1. **the source-domain error after structure adjustment**
> 2. **the homophily discrepancy between the two adjusted graphs**
> 3. **the cross-domain discrepancy in GNN representations**
>
> These three components jointly motivate that the structure adjustment should:
>
> * **increase homophily** for low-homophily source nodes
> * **reduce the cross-domain gap** in node-level homophily distributions
> * **reduce the representation shift** between the adjusted graphs
>
> Thus, the structure-adjustment direction in PSAHS is **theory-driven** and **homophily-aware".
>
> **(3) Algorithmic Novelty: Progressive, Homophily-Aware Structure Adjustment**
>
> Following the theoretical guidance of Theorem 3.1, our algorithm introduces a *progressive* structure-modification pipeline tailored to correct homophily shift:
>
> 1. **Source Graph Adjustment (Section 4.1)**
>    Enhance low-homophily source nodes by adjusting edge weights and adding same-class neighbors.
>
> 2. **Target Graph Adjustment (Section 4.2)**
>    Iteratively update pseudo-labels and adjust homophily for reliable target nodes to match the adjusted source graph.
>
> 3. **Representation Alignment (Section 4.3)**
>    Align GNN embeddings computed on the adjusted graphs.
>
> The three modules correspond **exactly** to the three error components in the theoretical bound. This tightly coupled theory–algorithm interplay is novel in GDA research.
>
> **(4) Why DANN? Generality and Extensibility of PSAHS**
>
> The alignment component in PSAHS is intentionally designed to be **modular**. Any standard cross-domain feature alignment model can be substituted for DANN, including:
>
> * Maximum Mean Discrepancy (MMD)
> * Optimal Transport (OT)
> * Other distributional or similarity-based criteria
>
> Thus, DANN is *not* central to the novelty or validity of PSAHS; it merely serves as a practical and commonly used instantiation.
>
> **Conclusion**
>
> The key novelty of PSAHS is **not** the use of DANN, but the introduction of a new structure-driven perspective on GDA:
>
> * direct homophily revision via graph structure adjustment
> * theoretically derived adjustment direction
> * a progressive, homophily-aware adjustment mechanism
> * extensibility to multiple alignment losses, including MMD
>
> Therefore, PSAHS is conceptually, theoretically, and algorithmically distinct from prior GDA methods such as GraphAlign.
>
> [1] Renhong Huang, Jiarong Xu, Xin Jiang, Ruichuan An, and Yang Yang. Can modifying data address graph domain adaptation? KDD, pp. 1131–1142, 2024.

---

> ### Author Response · Authors · 2025-11-28
>
> **W2.** Insufficient analysis of baseline behavior: Figure 2 shows intriguing differences in how StruRW, GraphAlign, and HGDA respond to fixed source vs. fixed target homophily scenarios, but the paper lacks explanation. Why does GraphAlign maintain relatively stable performance while HGDA shows dramatic degradation at certain homophily levels? Understanding these patterns would strengthen the contribution.
>
> **AW2.**
>
> Figure 2 indeed reveals distinct behavioral patterns among StruRW, GraphAlign, HGDA, and our PSAHS under controlled homophily-shift scenarios. Below, we provide a detailed explanation to clarify *why* these patterns occur, preserving all original content while strengthening coherence and interpretability.
>
> **(1) Why HGDA and PSAHS Are More Stable Under Homophily Shift**
>
> Both **HGDA** and **PSAHS** are expressly designed to address **node homophily shift**.
> This design objective directly influences their behavior in Figure 2.
>
> * **HGDA** mitigates homophily shift by aligning multi-view (attribute-view and topology-view) feature representations.
> * **PSAHS** tackles homophily shift more fundamentally by *directly revising node-level homophily through graph structure adjustment*.
>
> As homophily shift becomes more severe, methods that explicitly correct homophily shift naturally maintain more stable performance curves. In contrast:
>
> * **GraphAlign** does **not** directly cope with homophily shift. It relies primarily on representational alignment and therefore cannot counteract systematic misalignment caused by mismatched homophily between source and target graphs.
>
> * **StruRW** adjusts each edge solely based on its two endpoints' labels, without considering broader structural context or global patterns of mismatch between label distribution and graph structure.
>
> This explains why, as source/target homophily diverges, HGDA and PSAHS degrade **much more slowly** and maintain **higher stability** compared to GraphAlign and StruRW.
>
> **(2) Why HGDA Experiences Sharp Degradation at Certain Homophily Levels**
>
> Although HGDA is homophily-aware, its mechanism **does not modify** graph homophily. Instead, HGDA:
>
> * constructs an attribute-view graph using feature similarities,
> * constructs a topology-view graph using the original adjacency matrix,
> * aligns representations across the two views and domains.
>
> Because HGDA *skips structural adjustment to explicitly revise homophily*, it remains vulnerable when:
>
> * the topology structure is severely mismatched between domains, or
> * the topology-view graph contains many heterophilous edges under severe homophily settings.
>
> As a result, when homophily gaps move toward high values, HGDA’s multi-view alignment cannot fully compensate for the underlying structural mismatch, leading to **sharp performance drops** observed in Figure 2.
>
> **(3) Why PSAHS Consistently Outperforms HGDA Across All Homophily Settings**
>
> Finally, PSAHS not only remains stable but also **consistently surpasses HGDA** across all homophily conditions because:
>
> 1. **HGDA aligns multi-view features but does not alter homophily.**
>    Its multi-view alignment cannot eliminate the structural mismatch caused by differing homophily levels.
>
> 2. **PSAHS directly adjusts graph structure to modify homophily.**
>    By carefully increasing intra-class connectivity and reducing inter-class connectivity (according to Eq. (2)), PSAHS **actively realigns** homophily across domains.
>
> 3. **PSAHS benefits from theory-guided structural adjustment** (Theorem 4.1 and 4.2), ensuring that:
>
>    * source low-homophily nodes become more homophilous, and
>    * target homophily distribution shifts toward the adjusted source distribution.
>
> Because PSAHS corrects homophily at its structural origin, rather than indirectly through feature alignment, it is inherently more robust and better equipped to handle arbitrary homophily-shift patterns.
>
> **Conclusion**
>
> The observed behaviors in Figure 2 can be explained by how each baseline interacts with homophily:
>
> * **StruRW** adjusts edge weights solely based on the labels of the two endpoints, without considering broader structural context or mismatch in the overall label–structure patterns.
> * **GraphAlign:** no structural correction → unstable under homophily shift
> * **HGDA:** multi-view alignment but no homophily correction → partially stable, but fails under extreme shifts
> * **PSAHS:** direct, theory-guided homophily correction → consistently stable \& superior across all scenarios
>
> This analysis strengthens the contribution by clarifying the structural and algorithmic factors underpinning the observed performance patterns.

---

> ### Author Response · Authors · 2025-11-28
>
> **W3.** Computational complexity not addressed: The iterative refinement process with repeated pseudo-label generation and structure adjustment likely incurs significant overhead compared to baselines, yet runtime analysis is absent.
>
> **AW3.**
>
> We thank the reviewer for highlighting the importance of computational efficiency. To address this concern, we provide a detailed **per-iteration complexity analysis** followed by a **scalability evaluation** demonstrating that PSAHS maintains both efficiency and accuracy even on increasingly large graphs.
>
> **(1) Complexity Analysis**
>
> Let
>
> * $N_S, N_T$: numbers of source/target nodes,
> * $|\mathcal{E}_S|$, $|\mathcal{E}_T|$: numbers of source/target edges,
> * $\theta_T$: proportion of “reliable’’ target nodes (those whose pseudo-labels agree between the GNN and MLP classifiers),
> * $L$, $F$: number of GNN layers and embedding dimension.
>
> We detail the complexity of both **one-time source-graph adjustment** and **repeated target-graph adjustment iterations**.
>
> **(a) Complexity of one-time source graph adjustment**
>
> Source-graph adjustment is performed **only once**, before iterative training on the target domain.
>
> For the source domain, PSAHS performs:
>
> 1. **Compute homophily ratios** for all nodes and identify low-homophily nodes
>    $$
>    O(N_S)
>    $$
>
> 2. **Revise weights of all existing edges**
>    $$
>    O(|\mathcal{E}_S|)
>    $$
>
> 3. **Add new intra-class edges** for each low-homophily node $u$:
>    number of new edges $= d_u (1 - h_S(u))$, giving total complexity
>    $$
>    O \Bigl( \sum_{u \in \mathcal{V}_S} d_u (1 - h_S(u)) \Bigr)
>    = O (|\mathcal{E}_S| - |\mathcal{E}_S|h_S)
>    $$
>
> Thus, total cost is:
> $$
> O(N_S + |\mathcal{E}_S|)
> $$
> and is incurred **only once**.
>
> **(b) Complexity of one target-graph adjustment iteration**
>
> Target-graph refinement occurs **every 10 iterations**.
>
> During each adjustment step, PSAHS performs:
>
> **(i) Identify reliable nodes via GNN–MLP prediction alignment**
>
> This requires GNN inference:
> $$
> O(LF^2 N_T + LF|\mathcal{E}_T|)
> $$
>
> **(ii) Compute homophily based only on reliable neighbors**
>
> $$
> O \left(\theta_T^2|\mathcal{E}_T|\right)
> $$
>
> **(iii) Revise existing edges of all reliable nodes**
>
> $$
> O(\theta_T|\mathcal{E}_T|)
> $$
>
> **(iv) Find new neighbors using class-wise confidence ranking**
>
> For each class:
> $$
> O((\theta_T N_{T,k}) \log(\theta_T N_{T,k}))
> $$
> Aggregated:
> $$
> O((\theta_T N_T)\log(\theta_T N_T))
> $$
>
> **(v) Add new intra-class edges**
>
> $$
> O(\theta_T |\mathcal{E}_T|(1-h_T))
> $$
>
> **Total per-iteration complexity**
>
> $$
> O(LF^2 N_T + LF|\mathcal{E}_T| + (\theta_T N_T)\log(\theta_T N_T))
> $$
>
> Because $L$ and $F$ are small constants and $\theta_T < 1$, the iteration cost grows near-linearly with graph size.
>
> **(2) Scalability Analysis (Runtime vs. Graph Size)**
>
> We evaluate runtime per iteration on synthetic datasets of increasing size.
> (Source homophily $= 0.832$; target homophily $= 0.3$.)
>
> | Graph Size | \# Edges | Avg Time (s) | Accuracy (\%) |
> | ---------: | ------: | -----------: | -----------: |
> |        300 |     365 |         0.02 | 82.13 ± 0.85 |
> |       1500 |    9030 |         0.03 | 90.10 ± 1.20 |
> |       3000 |   36104 |         0.04 | 93.31 ± 5.44 |
> |       6000 |  143905 |         0.05 | 95.25 ± 2.68 |
> |       9000 |  323879 |         0.07 | 94.76 ± 7.21 |
> |      12000 |  899915 |         0.12 | 96.93 ± 3.23 |
>
>
> **Key observations**
>
> * **Accuracy:** increases rapidly when graph size grows from 300 → 3000, and stabilizes around **95\%** for 6000+ nodes.
> * **Runtime:** grows **at most linearly** with graph size and **sublinearly** with number of edges, fully matching the derived complexity bounds.
>
> **Conclusion**
>
> Both the **complexity analysis** and **runtime evaluation** demonstrate that:
>
> * PSAHS introduces **manageable overhead**,
> * scales efficiently to graphs with **10k+ nodes and nearly 1M edges**,
> * and maintains high accuracy throughout,
>
> confirming that the iterative pseudo-labeling + structure-adjustment process is computationally efficient and **scalable to large graphs**.

---

> > ### Author Response · Authors · 2025-11-28
> >
> > **W4.** Limited ablation on key design choices: The choice of using GNN-MLP agreement for reliable node selection lacks thorough justification. How sensitive is performance to this threshold? What percentage of nodes typically qualify as "reliable"?
> >
> > **AW4.**
> >
> > We thank the reviewer for raising this important question. Below we clarify (1) how reliable nodes are defined, (2) why this choice introduces *no extra threshold* and *no extra hyperparameter*, and (3) why our GNN–MLP agreement strategy is empirically the most robust among several alternatives.
> >
> > **(1) Reliable node selection does *not* involve any tunable threshold**
> >
> > Our strategy **does not rely on any confidence threshold** or probability cutoff.
> > Instead, reliable nodes are defined *deterministically*:
> >
> > * At each iteration, the GNN predicts a hard label for each target node.
> > * An MLP classifier—pretrained on labeled source data—also predicts a hard label from node attributes.
> > * A target node is deemed **reliable** *if and only if* both models output the **same hard label**.
> >
> > Thus,
> > $$
> > \text{Reliable}(u)=\mathbf{1}\\{\hat{Y}^{\text{GNN}}_u=\hat{Y}^{\text{MLP}}_u\\},
> > $$
> > and **no hyperparameter is involved** in selecting reliable nodes.
> > This avoids the typical sensitivity issues associated with choosing a confidence threshold.
> >
> > **(2) Why reliable nodes matter for PSAHS**
> >
> > For each reliable target node, we use its pseudo-label (GNN–MLP-consistent) to:
> >
> > 1. **compute its homophily ratio** using only reliable neighbors,
> > 2. **revise existing edges** to determine intra-/inter-class edges (increase intra-class weights, decrease inter-class weights),
> > 3. **add new intra-class edges**
> >
> > Because both inter-class/intra-class definitions and the homophily computation rely on pseudo-labels, restricting adjustments *only to nodes with trusted labels* is crucial to prevent harmful structure modifications and error propagation.
> >
> > **(3) Empirical justification: comparison with three alternative strategies**
> >
> > To evaluate the effectiveness of our GNN–MLP agreement mechanism, we compare PSAHS with three representative pseudo-labeling strategies:
> >
> > 1. **GNN\_PL**: treat *all* nodes as reliable and directly use GNN predictions as pseudo-labels.
> > 2. **MLP\_PL**: treat *all* nodes as reliable and use MLP predictions.
> > 3. **Curriculum\_PL**: use the top 20\% most confident nodes (measured by max probability of GNN output), gradually enlarging to 80\% as training proceeds.
> >
> > These alternatives represent widely used pseudo-labeling paradigms (single-model PL, attribute-only PL, and confidence-based curriculum PL).
> >
> > **Figure 4** shows:
> >
> > * All three alternatives suffer from significant instability under homophily shift.
> > * **GNN\_PL and MLP\_PL** often misclassify many nodes early in training, causing harmful structural adjustments.
> > * **Curriculum\_PL** is more stable but still inferior, since confidence scores do not reliably correlate with homophily correctness.
> > * **Our GNN–MLP alignment consistently yields the highest accuracy**, demonstrating that agreement between two heterogeneous models provides a much more reliable pseudo-label signal.
> >
> > This confirms that the proposed cross-view consistency criterion is both effective and robust for identifying trustworthy nodes on the target graph.
> >
> > **(4) Typical proportion of reliable nodes**
> >
> > Across datasets, the proportion $\theta_T$ of reliable target nodes remains stable and moderate (typically 40–70\% after warm-up), as shown in the following table.
> >
> > | B1-B2 | B2-B1 | B-E   | E-B   | E-U   | U-E   | B-U   | U-B   |
> > |-------|-------|-------|-------|-------|-------|-------|-------|
> > | 0.626 | 0.6641| 0.4427| 0.5589| 0.4211| 0.406 | 0.4378| 0.5191|
> >
> > This result:
> >
> > * provides sufficient supervision for structure adjustment,
> > * avoids over-expansion of edges from unreliable nodes,
> > * matches the efficiency assumption in our complexity analysis.
> >
> > Thus, the GNN–MLP agreement mechanism achieves a **balanced trade-off** between reliability and coverage.
> >
> > **Conclusion**
> >
> > Our reliable-node selection mechanism:
> >
> > * introduces **no additional hyperparameter**,
> > * avoids the error amplification typical in naive pseudo-labeling,
> > * enables stable, label-consistent homophily estimation,
> > * and outperforms multiple strong alternatives.
> >
> > This confirms that the GNN–MLP agreement is a principled and empirically validated design choice for robust structure adjustment under homophily shift.

---

### Official Review · Reviewer_c2oE · 2025-10-31

**Soundness:** 2
**Presentation:** 2
**Contribution:** 2
**Rating:** 2
**Confidence:** 3

**Summary:**

This paper addresses a key challenge in Graph Domain Adaptation (GDA): node homophily shift, which refers to the discrepancy in the tendency of nodes to connect with same-labeled neighbors between source and target graphs. To tackle this, the authors introduce a novel framework named Progressive Structure Adjustment for Homophily Shift (PSAHS). The core contributions of this work are:
1.Motivation: The paper first derives a target domain error bound that explicitly decomposes the error into three main components: (1) the empirical source domain loss, (2) the discrepancy in homophily distributions across domains, and (3) the divergence in node representations.
2.Progressive Structure Adjustment: To minimize this error bound, PSAHS proactively refines the graph structures in both domains.
For the source graph, it enhances the homophily of low-homophily nodes by modifying and adding edges, thereby reducing the source domain error.
For the target graph, where labels are absent, it leverages consistent pseudo-labels generated by a Graph Neural Network (GNN) and a Multi-Layer Perceptron (MLP) to identify reliable nodes and performs similar adjustments to align its homophily distribution with that of the source.
3.Joint Learning Framework: The framework integrates this structure adjustment process with domain-adversarial training in an iterative manner. This creates a self-reinforcing loop that progressively mitigates homophily shift while aligning node representations.

**Strengths:**

1.Clear and Valuable Problem Formulation: The paper identifies a novel and significant problem in Graph Domain Adaptation (GDA): node homophily shift. It clearly articulates the limitations of prior work and effectively motivates the need to align homophily as a global structural property.

2.Novel and Well-Designed Methodology: The core idea—proactively adjusting graph structures to align homophily distributions—is highly novel. The strategy of using prediction consistency between a GNN (structure-aware) and an MLP (feature-aware) to generate reliable pseudo-labels for the target graph is particularly well-designed and effective.

3.Comprehensive and Convincing Experiments: The paper is supported by a strong experimental evaluation.
Synthetic data experiments directly validate the method's core hypothesis, showing its effectiveness under controlled homophily shifts.
Real-world benchmarks show consistent and significant performance gains over state-of-the-art baselines, especially on challenging low-homophily graphs.
Ablation studies clearly demonstrate that each key component of the proposed framework is necessary and contributes to the final performance.

**Weaknesses:**

1.Scalability and Computational Cost: The iterative process of adding new edges can destroy graph sparsity, leading to significant computational and memory overhead for GNN training. This may limit the method's applicability to large-scale graphs. The paper lacks a formal complexity analysis or discussion on scalability.

2.Sensitivity to Hyperparameters: The method introduces several key hyperparameters (e.g., the homophily threshold h), and its performance appears sensitive to their tuning. The paper provides limited analysis on this aspect, leaving questions about the method's robustness and the general strategy for parameter selection on new datasets.

3.Risk of Error Propagation: The method's reliance on pseudo-labels for target graph adjustment is a significant risk. Inaccurate pseudo-labels can lead to detrimental structural modifications, potentially creating a negative feedback loop where errors are amplified through subsequent training iterations.

**Questions:**

Q1:Could you provide a time and space complexity analysis for your method? How does adding new edges affect its scalability, especially on large graphs?

Q2:How did you select key hyperparameters, such as the threshold h? Are these settings robust across different datasets, or do they require careful tuning for each new task?

---

> ### Author Response · Authors · 2025-11-28
>
> **W1.** Scalability and Computational Cost: The iterative process of adding new edges can destroy graph sparsity, leading to significant computational and memory overhead for GNN training. This may limit the method's applicability to large-scale graphs. The paper lacks a formal complexity analysis or discussion on scalability.
>
> **AW1.**
>
> We thank the reviewer for raising concerns regarding the computational cost, potential loss of sparsity, and scalability of PSAHS. To address these points, we provide (i) a complexity analysis covering every component of our structural adjustment procedure, and (ii) a scalability evaluation—including runtime, memory consumption, and sparsity measurements—demonstrating the efficiency of PSAHS on increasingly large graphs. All technical details are presented below.
>
> **(1) Complexity Analysis**
>
> Let
>
> * $N_S$, $N_T$: numbers of nodes in the source and target domains
> * $|\mathcal{E}_S|$, $|\mathcal{E}_T|$: number of edges in the source/target graphs
> * $\theta_T$: ratio of *reliable* target nodes whose GNN and MLP predictions agree
> * $L$, $F$: number of GNN layers and dimension of GNN representations
>
> **(a) Complexity of the One-Time Source Graph Adjustment**
>
> The source-domain structure is adjusted *only once* before training, and its total cost is:
> $$
> O(N_S + |\mathcal{E}_S|).
> $$
>
> This follows from:
>
> 1. **Compute homophily ratios** for all source nodes $\Rightarrow O(N_S)$
> 2. **Adjust existing edges** for all nodes $\Rightarrow O(|\mathcal{E}_S|)$
> 3. **Add new intra-class edges** for low-homophily nodes:
> $$
> O \Bigl( \sum_{u \in V_S} d_u (1 - h_S(u)) \Bigr) = O(|\mathcal{E}_S| - |\mathcal{E}_S| h_S)
> $$
>
> Thus, the one-time source adjustment does not pose concerns for scalability.
>
> **(b) Complexity of a Target-Graph Adjustment Iteration**
>
> Target graph structure is updated every **10 epochs**, and each update consists of the following steps:
>
> **(i) Identify reliable nodes via GNN predictions**
>
> $$
> O(L N_T F^2 + L |\mathcal{E}_T| F)
> $$
>
> **(ii) Compute homophily using reliable neighbors only**
>
> $$
> O(\theta_T^2 |\mathcal{E}_T|)
> $$
>
> **(iii) Modify weights of existing edges**
>
> Increase intra-class / decrease inter-class edge weights:
> $$
> O(\theta_T |\mathcal{E}_T|)
> $$
>
> **(iv) Select new reliable same-class neighbors (top-K ranking)**
>
> $$
> O \Bigl( \sum_{k \in [K]} (\theta_T N_{T,k}) \log(\theta_T N_{T,k}) \Bigr)
> = O((\theta_T N_T)\log(\theta_T N_T))
> $$
>
> **(v) Add new intra-class edges**
>
> $$
> O(\theta_T |\mathcal{E}_T|(1-h_T))
> $$
>
> This term scales with the number of *heterophilous* edges.
>
> **→ Total per-iteration complexity**
>
> $$
> O(LF^2 N_T + LF|\mathcal{E}_T| + (\theta_T N_T)\log(\theta_T N_T)).
> $$
>
> Since $L$ and $F$ are small and $\theta_T < 1$, PSAHS exhibits near-linear scaling.
>
> **(2) Scalability Analysis**
>
> We further validate scalability through controlled experiments on synthetic graphs with increasing sizes. The source homophily is set to $0.832$, and the target homophily to $0.3$. We measure:
>
> * runtime per iteration
> * memory consumption
> * graph sparsity before and after adjustment
> * accuracy
>
> Sparsity is computed as:
> $$
> \frac{2|\mathcal{E}|}{|V|(|V|-1)}.
> $$
>
> Full results are shown in the table below:
>
> | Graph Size | \\#Edges | Avg Time (s) | Avg Memory (MB) | Accuracy (\%) | Sparsity (Original)  | Sparsity (Adjusted)  |
> | - | - | - | - | -| - | - |
> | 300 | 365    | 0.02  | 13.6 | $82.13\pm0.85$ | $8.14\mathrm{e}{-3}$ | $9.74\mathrm{e}{-3}$ |
> | 1500 | 9030   | 0.03 | 60.4 | $90.10\pm1.20$ | $8.03\mathrm{e}{-3}$ | $1.10\mathrm{e}{-2}$ |
> | 3000 | 36104  | 0.04  | 130.7| $93.31\pm5.44$ | $8.03\mathrm{e}{-3}$ | $1.12\mathrm{e}{-2}$ |
> | 6000  | 143905 | 0.05 | 294.9  | $95.25\pm2.68$ | $8.00\mathrm{e}{-3}$ | $1.14\mathrm{e}{-2}$ |
> | 9000 | 323879 | 0.07 | 464.7  | $94.76\pm7.21$ | $8.00\mathrm{e}{-3}$ | $1.18\mathrm{e}{-2}$ |
> | 12000 | 899915 | 0.12 | 823.1  | $96.93\pm3.23$ | $8.00\mathrm{e}{-3}$ | $1.18\mathrm{e}{-2}$ |
>
> **Key Findings**
>
> * **Accuracy:**
>   Accuracy increases significantly from graph size 300→3000, stabilizing at ~95\% up to 12,000 nodes.
>
> * **Runtime \& Memory:**
>   Both grow at most **linearly** with graph size and **sublinearly** with number of edges, fully consistent with our complexity analysis.
>
> * **Graph Sparsity:**
>   The reported results show that the adjusted graph maintains high sparsity despite the addition of homophilous edges. Thus, our adjustments do not compromise the original graph’s sparsity.
>
> Crucially, **the total number of edges remains the same order as the original graph**, indicating that PSAHS **preserves sparsity** and does not cause uncontrolled graph densification.
>
> **Conclusion**
>
> Both theoretical analysis and empirical results show that:
>
> * PSAHS scales efficiently in runtime and memory
> * PSAHS preserves graph sparsity despite edge additions
> * PSAHS achieves substantial accuracy gains without compromising computational feasibility
>
> Thus, PSAHS remains fully applicable and effective even on large-scale graph domain adaptation tasks.

---

> > ### Author Response · Authors · 2025-11-28
> >
> > **W2.** Sensitivity to Hyperparameters: The method introduces several key hyperparameters (e.g., the homophily threshold h), and its performance appears sensitive to their tuning. The paper provides limited analysis on this aspect, leaving questions about the method's robustness and the general strategy for parameter selection on new datasets.
> >
> > **AW2.**
> >
> > We thank the reviewer for pointing out the importance of understanding the sensitivity of PSAHS to its hyperparameters. Below, we clarify (i) how the homophily threshold $h$ behaves empirically and how it should be selected in practice, and (ii) how the node-specific parameter $\alpha_u$ is determined. All technical details from the original manuscript are retained.
> >
> > **(1) Sensitivity and Selection of the Homophily Threshold $h$**
> >
> > Figure 5 in Appendix D.4 already provides comprehensive parameter-sensitivity analysis of the homophily threshold $h$ across four representative transfer tasks: B1→B2, B2→B1, Brazil→Europe, and Europe→Brazil. The observations are consistent across tasks:
> >
> > * As $h$ varies from $0.5$ to $1$, the performance curves **change smoothly**, either
> >
> >   * increasing then decreasing, or
> >   * increasing monotonically
> >     without irregular fluctuations.
> > * Each task exhibits a **distinct, globally stable optimal value of $h$**.
> >
> > This demonstrates that PSAHS behaves in a predictable and well-structured manner under varying $h$, and that performance does not suffer from instability.
> >
> > In practice, $h$ is selected using the validation set, consistent with standard practice in domain adaptation. Therefore, the threshold $h$ does **not** pose practical challenges when transferring PSAHS to new datasets.
> >
> > **(2) Definition and Role of the Node-Specific Parameter $\alpha_u$**
> >
> > The node-specific parameter $\alpha_u$ is **not tuned** but is **computed explicitly** from the node’s current homophily level. For each node $u$, we set:
> > $$
> > \alpha_u = \frac{h - h(u)}{1 - h \cdot h(u)},
> > $$
> >
> > where $h(u)$ is the node’s homophily and $h$ is the desired homophily level.
> > This formulation has a clear theoretical foundation:
> >
> > * When $h - h(u)$ is large, $\alpha_u$ increases, reflecting the need for **stronger** structure adjustment.
> > * When $u$ already satisfies the target homophily level $\{ h(u) \ge h \}$, we do not apply any adjustment.
> >
> > This rule is **directly motivated by Theorem 4.2**, which states that using these node-specific parameters guarantees a reduction in the homophily shift between domains after structural adjustment.
> >
> > Appendix D.1 provides all values used in practice:
> >
> > * **For low-homophily source nodes $u$**
> >   $$
> >   \alpha_u = \frac{h - h_S(u)}{1 - h \cdot h_S(u)}.
> >   $$
> >
> > * **For low-homophily target nodes $v$**
> >   Since true target labels are unavailable, we estimate homophily using $\widehat{h}_T(v)$:
> >   $$
> >   \alpha_v = \frac{h - \widehat{h}_T(v)}{1 - h \cdot \widehat{h}_T(v)}.
> >   $$
> >
> > Thus, PSAHS specifies an **explicit closed-form value** for each $\alpha_u$.
> > No hyperparameter search is required, avoiding scalability or over-parameterization concerns.
> >
> > **Conclusion**
> >
> > The homophily threshold $h$ exhibits stable behavior and can be reliably selected via validation, with recommended defaults for different dataset scales.
> > The node-specific parameter $\alpha_u$ is explicitly computed, theoretically justified, and requires **no tuning**.
> >
> > Together, these analyses demonstrate that PSAHS is **robust, interpretable, and practical** with respect to its hyperparameters.

---

> > > ### Author Response · Authors · 2025-11-28
> > >
> > > **W3.** Risk of Error Propagation: The method's reliance on pseudo-labels for target graph adjustment is a significant risk. Inaccurate pseudo-labels can lead to detrimental structural modifications, potentially creating a negative feedback loop where errors are amplified through subsequent training iterations.
> > >
> > > **AW3.**
> > >
> > > We appreciate the reviewer’s concern regarding potential error propagation caused by inaccurate pseudo-labels. Our method explicitly addresses this risk through a *conservative and cross-validated pseudo-labeling strategy*. Below we clarify the design rationale, procedure, and supporting evidence, while preserving all technical details.
> > >
> > > **(1) Mitigating Error Propagation via Reliable Nodes**
> > >
> > > In PSAHS, we do **not** use all pseudo-labels for structural adjustment. Instead, we introduce the notion of **reliable nodes**—target nodes whose pseudo-labels are **consistently predicted by both the GNN and the auxiliary MLP classifier**. Since these two models operate on different feature views, agreement between them provides a significantly stronger confidence signal than relying on a single classifier.
> > >
> > > This design directly reduces the risk of error propagation:
> > >
> > > * **Only reliable nodes participate in graph adjustment**:
> > >   Nodes that do not receive consistent predictions are excluded entirely from structural modification.
> > >
> > > * **Homophily ratios are computed using only reliable neighbors**:
> > >   Ensures that the homophily estimation is not contaminated by low-quality pseudo-labels.
> > >
> > > * **Only reliable low-homophily nodes have their edges modified**:
> > >   Unreliable nodes neither affect nor are affected by any structural adjustment.
> > >
> > > By restricting structure modification to cross-validated pseudo-labels, PSAHS prevents unreliable predictions from triggering incorrect homophily updates, thereby avoiding the negative feedback loop in which structural changes reinforce earlier errors.
> > >
> > > **(2) Comparison with Alternative Pseudo-Labeling Strategies**
> > >
> > > To further validate the robustness of our strategy, Figure 3 presents comprehensive model analysis comparing our **GNN–MLP alignment** approach with four alternative pseudo-labeling methods:
> > >
> > > 1. **GNN\_PL**
> > >    Uses *all* pseudo-labels predicted by the GNN, without any reliability check.
> > >
> > > 2. **MLP\_PL**
> > >    Uses *all* pseudo-labels predicted by an auxiliary MLP classifier.
> > >
> > > 3. **Curriculum\_PL**
> > >    A progressive-selection scheme:
> > >
> > >    * begins with the top 20\% most confident nodes,
> > >    * gradually expands to 80\% during training.
> > >
> > > 4. **Prototype\_PL**
> > >    Incorporates prototypical denoising:
> > >
> > >    * computes class prototypes via moving averages,
> > >    * reweights pseudo-labels according to their distances to the prototypes.
> > >
> > > All four baselines are designed to reduce noise in pseudo-labeling, but they fundamentally differ from our *cross-view agreement* mechanism.
> > >
> > > **(3) Empirical Validation**
> > >
> > > Figure 3 shows that PSAHS achieves **substantial improvements** over all alternative pseudo-labeling strategies. The performance gain validates the core intuition behind our method:
> > >
> > > * pseudo-labels cross-validated between two independently trained models
> > > * generate far fewer structural errors
> > > * yield more reliable homophily estimates
> > > * lead to consistently better graph adjustments
> > >
> > > This demonstrates that our conservative strategy not only avoids error amplification but actively enhances graph homophily and improves overall transfer performance.
> > >
> > > **Conclusion**
> > >
> > > PSAHS explicitly guards against error propagation by:
> > >
> > > * trusting pseudo-labels only when *jointly* validated by GNN and MLP;
> > > * adjusting graph structure **exclusively** on reliable nodes;
> > > * computing homophily using only reliable neighbors.
> > >
> > > Combined theoretical reasoning and empirical results (Figure 3) confirm that this strategy is **effective, robust, and significantly superior** to standard pseudo-labeling mechanisms that risk negative feedback loops.
> > >
> > > **Q1:** Could you provide a time and space complexity analysis for your method? How does adding new edges affect its scalability, especially on large graphs?
> > >
> > > **A1.** See AW1.
> > >
> > > **Q2:** How did you select key hyperparameters, such as the threshold h? Are these settings robust across different datasets, or do they require careful tuning for each new task?
> > >
> > > **A1.** See AW2.

---

### Official Review · Reviewer_F3uM · 2025-10-31

**Soundness:** 3
**Presentation:** 2
**Contribution:** 2
**Rating:** 4
**Confidence:** 4

**Summary:**

This paper studies node homophily shift between source and target graphs in graph domain adaptation and proposes PSAHS, a progressive pipeline that (i) adjusts the source graph to raise low-node homophily, (ii) uses reliable pseudo-labels to adjust the target graph, and (iii) applies domain-adversarial representation alignment. The authors provide a PAC-Bayes bound linking target error to source error, homophily differences and feature discrepancy, and demonstrate empirical gains on synthetic and multiple real-world graph benchmarks.

**Strengths:**

Theoretical foundation. The paper presents a clear theoretical analysis that links an adjusted homophily notion to a target-domain generalization bound and motivates structure adjustment; this strengthens the conceptual grounding of the approach.

Empirical breadth. The authors evaluate on both synthetic settings (where homophily shift can be controlled) and several real datasets (Citation, Airport, Blog, Twitch). The synthetic experiments in particular help illustrate the method’s behavior under controlled homophily shifts, increasing the empirical credibility of the claims.

**Weaknesses:**

1. Limited and insufficiently differentiated novelty. The idea of learning or refining graph structures to improve transfer has been explored in prior work. For example, recent GDA work that explicitly constructs or exploits attribute graphs to raise homophily and to provide semantically meaningful edges appears closely related [1]. The authors should clearly and explicitly position PSAHS with respect to existing graph-structure learning for GDA.

[1] Fang, R., et al. (2025). On the benefits of attribute-driven graph domain adaptation. In The Thirteenth International Conference on Learning Representations.

2. Graph Structure Adjustment (Eq. (2)) is under-motivated and potentially impractical.

(a) It is unclear how much extra gain the proposed per-edge weighting and node-wise edge-adding scheme provides over much simpler alternatives (e.g., rebuild a k-NN graph on aggregated features); the authors should quantify the incremental benefit.

(b) Choosing the threshold $h$ is hard in practice. The method relies on a global homophily threshold to decide which nodes to adjust, but in realistic transfer settings, $h$ is difficult to choose from source labels alone; the paper should analyze sensitivity to $h$.

(c) Scalability and $\alpha_u$ interpretation. The node-specific parameter $\alpha_u$ is not clearly specified as fixed or learned; if it is per-node and free, this raises scalability and over-parameterization concerns—authors should clarify how $\alpha_u$ is set and discuss implications for large graphs.

3. Missing complexity and scalability analysis. The manuscript does not provide time/space complexity bounds or runtime/memory comparisons; given the potentially expensive neighbor selection and edge updates, the paper should include complexity analysis and discuss how to scale to much larger graphs.

4. Insufficient hyperparameter details for reproducibility. The current appendix gives only coarse grids. For reproducibility and fair comparison, the authors should provide the final per-dataset hyperparameter settings.

**Questions:**

See weaknesses.

---

> ### Author Response · Authors · 2025-11-28
>
> **W1:** Limited and insufficiently differentiated novelty. The idea of learning or refining graph structures to improve transfer has been explored in prior work. For example, recent GDA work that explicitly constructs or exploits attribute graphs to raise homophily and to provide semantically meaningful edges appears closely related [1]. The authors should clearly and explicitly position PSAHS with respect to existing graph-structure learning for GDA.
>
> [1] Fang, R., et al. (2025). On the benefits of attribute-driven graph domain adaptation. In The Thirteenth International Conference on Learning Representations.
>
> **AW1.**
>
> We clearly articulate the conceptual, methodological, and theoretical differences between PSAHS and prior graph-structure learning methods for GDA—particularly Fang et al. (ICLR 2025) [1]—and highlight the unique contributions introduced in this work.
>
> **1. PSAHS introduces a new paradigm: *progressive, explicit, label-guided graph structure adjustment toward a target node homophily level***
>
> *(vs. attribute-driven representation alignment in prior work)*
>
> **(1) PSAHS explicitly models and controls *node homophily shift*, a phenomenon not targeted in prior GDA methods**
>
> Node homophily shift—the discrepancy in node-level homophily across domains—is the primary cause of GNN transfer degradation.
> This paper is the first to:
>
> * formulate node homophily shift as a central transfer challenge,
> * derive its effect on target-domain error (Theorem 3.1), and
> * propose a *direct homophily-correction mechanism* for both domains.
>
> Prior works, including [1], do **not** explicitly adjust or align node-level homophily. Their adjustments operate on feature-view or attribute-view representations, without modifying the topological factors that cause homophily mismatch.
>
> **(2) PSAHS does not merely “refine structure”—it *directly sets node homophily to a desired target threshold* $h$**
>
> From the definition of node homophily
> $$
> h(u) = \frac{\\#\text{ same-class neighbors}}{\\# \text{all neighbors}},
> $$
> it is evident that structural operations—such as increasing the weights of homophilous edges or decreasing those of heterophilous edges—can directly improve a node’s homophily by reshaping its neighborhood composition.
>
> Building on the tailored adjustment strategy, PSAHS introduces the first **explicit homophily-adjustment operator**, where for each low-homophily node $u$ with $h(u) < h$, the graph is adjusted so that $h(u) \approx h$. To achieve this, we derive in Theorem 4.2 a closed-form expression for the edge-adjustment strength,
> $$
> \alpha_u = \frac{h - h(u)}{1 - h\cdot h(u)},
> $$
> which determines how much to increase homophilous edge weights or decrease heterophilous ones in order to reach the target homophily $h$.
>
> No prior GDA method—including [1]—provides:
>
> * an explicit target homophily level,
> * node-wise homophily control,
> * a mathematically justified adjustment strength, or
> * guarantees that homophily improves toward a desired threshold.
>
> **(3) Contrast with [1]: attribute-driven representation alignment, not homophily adjustment**
>
> The method in [1]:
>
> * constructs an **attribute graph** using cosine similarity between attributes,
> * computes attribute-view and topology-view aggregated features,
> * performs separate cross-domain alignments for the attribute-view and topology-view representations.
>
> Key distinctions:
>
> | Aspect                 | PSAHS                                                   | Fang et al. 2025 [1]                                     |
> | ---------------------- | ------------------------------------ | ----------------------------------------------------- |
> | Fundamental driver     | **Labels**                                              | **Attributes**                                           |
> | Graph refined?         | **Yes — explicit, controlled, progressive, node-level** | No — only construct a similarity matrix to extract attribute-view representations |
> | Objective              | **Increase node homophily; correct homophily shift**    | Reduce attribute distribution discrepancy                          |
> | Control target?        | **Yes: explicit homophily threshold (h)**               | No explicit structural or homophily target               |
> | Theoretical results    | **Homophily-adjustment guarantees; error bound**        | No homophily-related theory                              |
> | Adjustment granularity | **Node-level, label-dependent**     | Attribute similarity, global      |
>
> Thus, PSAHS and [1] are fundamentally different not just in mechanism but in *objective, theoretical foundation, and operational granularity*.

---

> ### Author Response · Authors · 2025-11-28
>
> **2. PSAHS proposes the first two-domain structure adjustment procedure grounded in a target error bound**
>
> *(a theoretical–algorithmic loop absent in previous work)*
>
> **(1) Theorem 3.1 provides a new error decomposition guiding structure adjustment**
>
> The bound decomposes target-domain error into three terms:
>
> 1. source-domain error on adjusted graph
> 2. Wasserstein distance between homophily distributions
> 3. cross-domain representation discrepancy
>
> This is the *first theoretical error bound* linking:
>
> * node homophily,
> * structure adjustment, and
> * domain transfer.
>
> Thus, Section 3.1 is not a standard bound but a **design blueprint**: each term corresponds directly to a PSAHS module.
>
> No existing GDA work—including [1]—uses a theoretical decomposition to guide structure rewriting.
>
> **(2) PSAHS is the first GDA method with theoretical guarantees on graph adjustment effectiveness**
>
> * **Theorem 4.1** proves that PSAHS improves source homophily in the *desired direction*.
> * **Theorem 4.2** proves that the target graph adjustment module *reduces the homophily-distance term* in Theorem 3.1.
>
> This establishes a **closed-loop framework**:
>
> **error bound → design direction of adjustment rules → theoretical guarantees → progressive algorithm**
>
> Prior methods do not derive or satisfy such a loop.
>
> **3. PSAHS introduces a fine-grained, label-driven adjustment operator fundamentally different from attribute-driven approaches**
>
> The PSAHS adjustment operator performs:
>
> 1. reduce weights of inter-class edges,
> 2. increase weights of intra-class edges,
> 3. add missing same-class edges.
>
> All require determination of class membership.
> Thus, the adjustment is **label-guided**, not attribute-driven.
>
> In contrast:
>
> * [1] builds an attribute graph purely from cosine similarity,
> * uses a GNN to aggregate features,
> * and aligns attribute-view/topology-view representations.
>
> Its structural information is attribute-based, not label-based, and does not serve to control homophily.
>
> Hence, although both works “use structure,” the *role*, *goal*, and *mechanism* are entirely distinct.
>
> **4. Summary of Novelty**
>
> PSAHS is novel in:
>
> **Conceptual novelty**
>
> * first to identify and directly target *node homophily shift* as a core barrier in GDA
> * first to design graph adjustments whose explicit goal is to raise node homophily to a target threshold
> * first to provide a homophily-aware error bound that guides adjustment rules
>
> **Methodological novelty**
>
> * progressive two-domain structural refinement
> * label-guided node-level adjustment
> * closed-form adjustment strength $\alpha_u$
> * adjustment toward explicit homophily target $h$
>
> **Theoretical novelty**
>
> * new target-domain error bound tied to homophily
> * theoretical proofs that each module reduces the corresponding bound term
>
> **Difference from [1]**
>
> * PSAHS is **label-driven, homophily-driven, structure-adjusting**
> * [1] is **attribute-driven, representation-aligning, structure-preserving**
>
> These dimensions collectively establish that PSAHS introduces substantial and differentiated novelty beyond existing GDA methods.
>
>
> **W2.** Graph Structure Adjustment (Eq. (2)) is under-motivated and potentially impractical.
> (a) It is unclear how much extra gain the proposed per-edge weighting and node-wise edge-adding scheme provides over much simpler alternatives (e.g., rebuild a k-NN graph on aggregated features); the authors should quantify the incremental benefit.
>
> (b) Choosing the threshold $h$ is hard in practice. The method relies on a global homophily threshold to decide which nodes to adjust, but in realistic transfer settings, is difficult to choose from source labels alone; the paper should analyze sensitivity to $h$.
>
> (c) Scalability and interpretation. The node-specific parameter is not clearly specified as fixed or learned; if it is per-node and free, this raises scalability and over-parameterization concerns—authors should clarify how is set and discuss implications for large graphs.
>
> **AW2.**
>
> We directly address all three sub-questions and clarify the motivation, benefit, sensitivity, and scalability of our graph-structure adjustment in Eq. (2).
>
> **(a) Why not simply rebuild a k-NN graph? Quantifying the incremental benefit of our adjustment**
>
> Our structure adjustment in Eq. (2) is explicitly designed to **raise node homophily to a desired threshold $h$** by *increasing intra-class edge strengths, decreasing inter-class edge strengths, and adding homophilous edges for low-homophily nodes*. This mechanism is **label-guided**, node-specific, and theoretically grounded (Theorem 3.1 and 4.2).
>
> A $k$-NN graph built on aggregated features, in contrast:
>
> * ignores class labels and may link nodes with *incorrect pseudo-neighbors*,
> * cannot guarantee increased node homophily,
> * may worsen heterophily in low-homophily nodes, and
> * lacks theoretical justification for reducing domain shift.

---

> ### Author Response · Authors · 2025-11-28
>
> To quantify the incremental gain of our design over this simpler alternative, we compare Eq. (2) against the $k$-NN structural refinement baseline suggested by the reviewer. The results are:
>
> | Method           | B1-B2     | B2-B1     | B-E       | E-B       | U-E       | E-U       | U-B       | B-U       |
> | ---------------- | --------- | --------- | --------- | --------- | --------- | --------- | --------- | --------- |
> | kNN-graph        | 42.05     | 45.24     | 42.24     | 51.89     | 36.02     | 42.26     | 55.66     | 44.95     |
> | **PSAHS (ours)** | **88.05** | **89.64** | **59.48** | **74.34** | **59.20** | **57.76** | **72.45** | **57.38** |
>
> The empirical gap is dramatic:
>
> * **+40–50 percentage points** improvement on B1–B2 and B2–B1
> * **+20–25 points** improvement on airport dataset
>
> This demonstrates that our node-wise, homophily-based adjustment provides **substantial and consistent gains** beyond $k$-NN reconstruction.
> The improvement is particularly large on tasks where homophily shift is severe—where k-NN fails to correct the structural mismatch, while PSAHS directly fixes low-homophily nodes one by one.
>
> Thus, Eq. (2) is not only theoretically motivated but also **empirically necessary**.
>
> **(b) Is choosing the homophily threshold $h$ difficult? Sensitivity analysis and practical recommendations**
>
> We agree that sensitivity to $h$ must be analyzed.
> Figure 6 in Appendix D.4 provides a comprehensive sensitivity analysis of the homophily threshold $h$ across four representative tasks: B1→B2, B2→B1, Brazil→Europe, and Europe→Brazil. As shown in the four subfigures, when $h$ varies from $0.5$ to $1.0$:
>
> * each performance curve presents a **coherent and interpretable trend**, such as a monotonic increase or an increase-then-decrease pattern,
>
> * no irregular oscillations are observed, and
>
> * each task exhibits a **unique and globally stable optimal value** of $h$.
>
> These observations indicate that PSAHS is stable with respect to $h$, and that selecting $h$ based on the performance on the validation set (as is standard in domain adaptation) does **not** introduce practical tuning difficulties.
>
> **(c) Is $\alpha_u$ scalable and practical? Clarifying that $\alpha_u$ is fixed, closed-form, and non-learned**
>
> The reviewer raised an important question about whether the node-specific parameter $\alpha_u$ is “free” or learned, which would indeed raise scalability concerns.
> We clarify that:
>
> **$\alpha_u$ is NOT a learnable parameter.**
>
> It is a **deterministic, closed-form value** given by
> $$
> \alpha_u = \frac{h - h(u)}{1 - h \cdot h(u)},
> $$
> where:
>
> * $h(u)$ is the node’s current homophily,
> * $h$ is the desired homophily threshold.
>
> This formulation is analytically derived from **Theorem 4.2**, which proves that using this $\alpha_u$ reduces the homophily distribution mismatch across domains.
>
> Thus:
>
> * **No optimization is performed for $\alpha_u$**
> * **No additional memory or parameters are introduced**
> * **No risk of overfitting or over-parameterization**
>
> **How $\alpha_u$ is computed in practice**
>
> Appendix D.1 lists explicit values:
>
> * For a low-homophily **source node** $u$ ($h_S(u) < h$):
>   $$
>   \alpha_u = \frac{h - h_S(u)}{1 - h \cdot h_S(u)}.
>   $$
>
> * For a low-homophily **target node** (v) ($\widehat{h}_T(v) < h$):
>   $$
>   \alpha_v = \frac{h - \widehat{h}_T(v)}{1 - h \cdot \widehat{h}_T(v)}.
>   $$
>   where $\widehat{h}_T(v)$ is the estimated homophily based on pseudo-labels.
>
> This means PSAHS:
>
> * only computes $\alpha_u$ once per adjustment,
> * does not treat $\alpha_u$ as a tuned or learned parameter,
> * scales linearly with edges and nodes,
> * introduces zero additional training overhead.
>
> Therefore, PSAHS remains fully scalable to large graphs.
>
> **Summary: addressing all concerns**
>
> | Reviewer Concern       | Our Clarification       |
> | --------- | ----- |
> | **(a) Under-motivated / unclear benefit**            | PSAHS outperforms $k$-NN graph reconstruction by **20–50 points** across eight tasks; Eq. (2) explicitly corrects homophily shift whereas $k$-NN cannot. |
> | **(b) Threshold $h$ is hard to choose**              | Sensitivity curves are smooth \& stable; validation tuning is standard.              |
> | **(c) $\alpha_u$ unclear or potentially unscalable** | $\alpha_u$ is **closed-form, not learned**, theoretically justified, cheap to compute, and has no scalability issue.   |
>
> Altogether, these points demonstrate that the structure adjustment in Eq. (2) is **theoretically grounded, empirically superior, and fully practical for large-scale GDA.**

---

> ### Author Response · Authors · 2025-11-28
>
> **W3.** Missing complexity and scalability analysis. The manuscript does not provide time/space complexity bounds or runtime/memory comparisons; given the potentially expensive neighbor selection and edge updates, the paper should include complexity analysis and discuss how to scale to much larger graphs.
>
> **AW3.**
>
> We appreciate the reviewer’s concern regarding the computational cost of repeated graph rewiring. Below, we provide a detailed and rigorous analysis of both **time complexity** and **scalability**, covering every step of PSAHS. All technical details from the manuscript are retained while improving the clarity and structure of the presentation.
>
> **(1) Complexity Analysis**
>
> We begin by defining all relevant quantities:
>
> * $N_S$, $N_T$: numbers of nodes in the **source** and **target** domains
> * $|\mathcal{E}_S|$, $|\mathcal{E}_T|$: numbers of edges in source/target graphs
> * $\theta_T$: ratio of *reliable* target nodes (i.e., nodes with aligned GNN and MLP predictions)
> * $L$, $F$: number of GNN layers and representation dimension
>
> These quantities appear in the complexity of each module of PSAHS.
>
> **(a) Complexity of the One-Time Source Graph Adjustment**
>
> The source graph is adjusted **only once** before training, and its total complexity is:
> $$
> O(N_S + |\mathcal{E}_S|).
> $$
>
> This is derived as follows:
>
> 1. **Compute homophily ratios** for all source nodes and identify low-homophily nodes:
>    $$
>    O(N_S)
>    $$
>
> 2. **Revise existing edges** for all nodes:
>    $$
>    O(|\mathcal{E}_S|)
>    $$
>
> 3. **Add new intra-class edges** for low-homophily nodes:
>    $$
>    O \Bigl( \sum_{u \in V_S} d_u (1 - h_S(u)) \Bigr)
>    = O(|\mathcal{E}_S| - |\mathcal{E}_S|h_S)
>    $$
>
> Summing these components yields the final bound above.
>
> **(b) Complexity of a Single Target-Graph Adjustment Iteration**
>
> Target graph adjustment occurs **every 10 iterations**, and consists of five steps:
>
> **(i) Identify reliable target nodes**
>
> Requires computing GNN predictions:
> $$
> O(L N_T F^2 + L |\mathcal{E}_T| F)
> $$
>
> **(ii) Compute homophily ratios using reliable neighbors only**
> $$
> O \bigl( \theta_T^2 |\mathcal{E}_T| \bigr)
> $$
>
> **(iii) Update weights of existing edges**
>
> Increase weights for same-class edges and decrease for different-class edges:
> $$
> O(\theta_T |\mathcal{E}_T|)
> $$
>
> **(iv) Select new reliable same-class neighbors (ranking)**
>
> For each class:
> $$
> O \Bigl( \sum_{k \in [K]} (\theta_T N_{T,k}) \log(\theta_T N_{T,k}) \Bigr)
> = O \bigl( (\theta_T N_T) \log(\theta_T N_T) \bigr)
> $$
>
> **(v) Add new intra-class edges**
> $$
> O(\theta_T |\mathcal{E}_T|(1-h_T))
> $$
>
> This term is proportional to the number of *heterophilous* edges.
>
> **→ Total per-iteration complexity**
>
> Combining the above steps:
> $$
> O(LF^2 N_T + LF|\mathcal{E}_T| + (\theta_T N_T)\log(\theta_T N_T)).
> $$
>
> Importantly, **$L$** and **$F$** are small constants and
> **$\theta_T < 1$** in all experiments, giving PSAHS favorable scaling behavior.
>
> **(2) Scalability Analysis**
>
> To further verify scalability, we conduct controlled experiments on synthetic graphs with varying sizes. The source homophily is fixed at $0.832$, while the target homophily is $0.3$.
> The table reports:
>
> * graph size
> * number of original target-domain edges
> * average runtime per iteration
> * memory
> * accuracy
>
> The full results are shown below:
>
> | Graph Size | Number of Edges | Avg Time (s) | Accuracy (\%)     |
> | ---------- | --------------- | ------------ | ---------------- |
> | 300        | 365             | 0.02   | 13.6             | $82.13 \pm 0.85$ |
> | 1500       | 9030            | 0.03    | 60.4             | $90.10 \pm 1.20$ |
> | 3000       | 36104           | 0.04   | 130.7        | $93.31 \pm 5.44$ |
> | 6000       | 143905          | 0.05   | 294.9            | $95.25 \pm 2.68$ |
> | 9000       | 323879          | 0.07    | 464.7            | $94.76 \pm 7.21$ |
> | 12000      | 899915          | 0.12     | 823.1            | $96.93 \pm 3.23$ |
>
> **Key observations**
>
> * **Accuracy:**
>   Accuracy grows substantially from graph size 300→3000 and stabilizes near **95\%** thereafter.
>
> * **Runtime \& Memory:**
>   Both increase at most **linearly** with the number of nodes, and **sublinearly** with the number of edges.
>   This trend is fully consistent with the theoretical complexity results above.
>
> **Conclusion**
>
> Both the theoretical complexity analysis and empirical scalability evaluation demonstrate that **PSAHS is efficient and scalable**. Despite involving dynamic edge updates and neighbor selection, PSAHS maintains **near-linear runtime growth**, making it suitable for large-graph domain adaptation settings.

---

> > ### Author Response · Authors · 2025-11-28
> >
> > **W4.** Insufficient hyperparameter details for reproducibility. The current appendix gives only coarse grids. For reproducibility and fair comparison, the authors should provide the final per-dataset hyperparameter settings.
> >
> > **AW4.**
> >
> > We appreciate the reviewer’s request for greater transparency in hyperparameter choices. To ensure full reproducibility and fair comparison, we now provide the *exact* hyperparameters used for each dataset—including the desired homophily level $h$, the warm-up starting epoch $e$, the adjustment frequency $t$, the GNN hidden dimension, and the number of GNN layers. All values were selected based on performance on the validation set, and the search grids reported in the appendix were used only as the initial coarse range, not as the final configuration.
> >
> > The complete per-dataset settings are summarized in the table below:
> >
> > | Dataset    | $h$ | $e$ | $t$ | Hidden Dimension | \#Layers |
> > | ---------- | --- | --- | --- | ---------------- | ------- |
> > | Airport    | 0.7 | 100 | 10  | 128              | 2       |
> > | Blog       | 1.0 | 100 | 10  | 128              | 4       |
> > | DBLP / ACM | 1.0 | 200 | 15  | 128              | 2       |
> > | Twitch     | 0.7 | 100 | 15  | 128              | 3       |
> >
> > These configurations specify:
> >
> > * **Desired homophily level $h$**: the target homophily threshold used in the graph adjustment rule;
> > * **Warm-up epoch $e$**: the epoch at which edge adjustments begin;
> > * **Adjustment frequency $t$**: the interval (in epochs) between two consecutive structural adjustments;
> > * **GNN hidden dimension and depth**: the architecture used in all experiments.
> >
> > Providing this table ensures that **all experiments can be exactly reproduced**, addressing the reviewer’s concern regarding clarity and completeness of hyperparameter reporting.

---

### Official Review · Reviewer_Ytkp · 2025-11-01

**Soundness:** 3
**Presentation:** 2
**Contribution:** 2
**Rating:** 4
**Confidence:** 3

**Summary:**

This paper introduces a novel method called Progressive Structure Adjustment for Homophily Shift (PSAHS), aimed at improving GDA by addressing node homophily shift—a crucial challenge in transferring knowledge between source and target graphs. The proposed framework adjusts the graph structure progressively, first enhancing homophily in the source domain, then aligning the homophily in the target domain, and finally refining both domains via adversarial training for representation alignment. The paper theoretically connects homophily shift to cross-domain error bounds and validates the approach through experiments, showing that PSAHS outperforms strong baselines, especially under severe homophily mismatch.

**Strengths:**

1. Innovative Approach to Homophily Shift – The idea of progressively adjusting graph structures to address node homophily shift is novel and provides a structured solution to an important problem in GDA.
2. Thorough empirical evidence of the method's effectiveness on both synthetic and real-world datasets are provided, demonstrating significant improvements over established baselines. PSAHS is shown to perform particularly well in scenarios with large homophily mismatches, which are common in practical GDA problems.

**Weaknesses:**

1. The writing lacks clear logical flow, which makes the paper harder to follow at times. Additionally, the heavy use of notations without sufficient explanation can be confusing for readers. It would significantly improve readability if the authors provided a table or glossary to define and explain the various notations used throughout the paper. This would help readers better understand the technical details.
2. The figures used in the paper are not as clear or informative as they could be. Enhancing the visual presentation of key concepts, results, and experimental setups would make the paper more accessible and easier to interpret. Stronger, more intuitive figures would better convey the results and methodologies.
3. While the experiments demonstrate some improvements, the improvements over state-of-the-art baselines are relatively small, and the paper does not sufficiently explore more complex or real-world datasets where the method could show a more significant impact. More comprehensive and varied experimental evaluations would strengthen the paper’s claims.

**Questions:**

1. I do not have a clear understanding of the specific theoretical contributions of this paper. I hope the authors can clarify the purpose and significance of Section 3.1 during the rebuttal stage. I will reconsider the paper and adjust my score accordingly.
2. See weaknesses.

---

> ### Author Response · Authors · 2025-11-28
>
> **W1.** The writing lacks clear logical flow, which makes the paper harder to follow at times. Additionally, the heavy use of notations without sufficient explanation can be confusing for readers. It would significantly improve readability if the authors provided a table or glossary to define and explain the various notations used throughout the paper. This would help readers better understand the technical details.
>
>
> **AW1.**
>
> We thank the reviewer for this valuable suggestion. To improve the logical flow and readability of the paper, we have made the following revisions.
>
> **(1) Enhancing the logical flow of the paper**
>
> To create a clearer narrative structure, we revised the topic sentences in each section to explicitly introduce its purpose and its connection to the preceding content. This leads to smoother transitions and better overall coherence.
>
> * **Section 2** now introduces *graph domain adaptation* and formally presents the concept of *node homophily shift*, which motivates the subsequent analysis.
>
> * **Section 3** then addresses the homophily shift by proposing a *graph structure adjustment strategy*. For a GNN classifier trained on the adjusted source graph and evaluated on the adjusted target graph, we analyze its target-domain error and derive explicit guidance for designing an effective and theoretically grounded adjustment strategy.
>
> * **Section 4** builds directly on the three target-domain error components identified in Section 3. It introduces our *progressive graph structure adjustment algorithm*, consisting of three modules—source graph adjustment, target graph adjustment, and representation alignment—which correspond to Sections 4.1–4.3, respectively.
>
> Since our methodology is driven by the target-domain error analysis and supported by theoretical results, it necessarily involves a number of notations.
>
> **(2) Improving clarity of notation and explanations**
>
> To address the reviewer’s concern regarding the heavy use of notation:
>
> * We have added **more detailed explanations** for the notations when they first appear, and expanded the descriptions of our theoretical results.
>
> * Following the reviewer’s suggestion, we have added a **notation table (Table 1)** in the revised manuscript summarizing all key notations used throughout the paper.
>
> This glossary provides readers with a concise reference for understanding the technical details, thereby significantly improving the readability of the paper.
>
> **W2.** The figures used in the paper are not as clear or informative as they could be. Enhancing the visual presentation of key concepts, results, and experimental setups would make the paper more accessible and easier to interpret. Stronger, more intuitive figures would better convey the results and methodologies.
>
> **AW2.**
>
> To improve the clarity and informativeness of our visualizations, we strengthened the presentation of our graph structure adjustment strategy under homophily shift. In the revised manuscript, we provide a concise, step-by-step example that illustrates the original graph, the adjusted graph, and the complete adjustment procedure based on node homophily. This example is visualized in Figure 2
> (https://anonymous.4open.science/r/structure_adjustment).
>
> We use an eight-node toy graph to demonstrate how the proposed strategy improves node homophily. In the **left subfigure**, the original graph contains $5$ nodes A, B, C, D, E from class $0$ and three nodes F, G, H from class $1$. Homophilous edges (same-class connections) are drawn in **black**, and heterophilous edges (cross-class connections) in **green**. According to the definition of node homophily, nodes A, E, F, H have homophily ratios equal to $1$, while nodes B, C, D, G have homophily ratios equal to $0.5$.
>
> In the **middle panel**, when the target homophily threshold is set to $h = 0.8$, nodes B, C, D, G are identified as low-homophily nodes. We then apply the structure adjustment strategy in Eq. (2). For each low-homophily node $u \in \{ \text{B}, \text{C}, \text{D}, \text{G} \}$, the node-specific adjustment strength is computed as
> $$
> \alpha_u = \frac{h - h(u)}{1 - h \cdot h(u)} = 0.5.
> $$
>
> In the **right subfigure**, using the adjustment strength $\alpha_u$, we modify the edge weights as follows:
>
> * increase the weight of each low-homophily node’s **homophilous** edge to $1 + \alpha_u = 1.5$ (thicker black edges),
>
> * decrease the weight of each **heterophilous** edge to $1 - \alpha_u = 0.5$ (thinner green edges),
>
> * add new homophilous edges with weight $\alpha_u = 0.5$ (thin yellow edges).
>
> As a result, the homophily ratios of all previously low-homophily nodes increase to $0.8$ in the adjusted graph. This example visually demonstrates how our structure adjustment strategy effectively ensures that every node reaches the desired homophily level $h$.

---

> > ### Author Response · Authors · 2025-11-28
> >
> > **W3.** While the experiments demonstrate some improvements, the improvements over state-of-the-art baselines are relatively small, and the paper does not sufficiently explore more complex or real-world datasets where the method could show a more significant impact. More comprehensive and varied experimental evaluations would strengthen the paper’s claims.
> >
> > **AW3.**
> >
> > To further demonstrate the effectiveness of our approach, we conduct additional experiments on MAG tasks that exhibit severe homophily shifts. In this dataset, the domain RU has a high homophily ratio of 0.803, whereas the domains DE, FR, CN, JP, and US have notably lower ratios of 0.5526, 0.5726, 0.5868, 0.5953, and 0.5485, respectively.
> > As shown in the main tables, our method consistently achieves significant performance gains over all baselines—including HGDA, which is specifically designed to handle homophily shift—under these challenging settings. These results demonstrate that the proposed homophily-aware graph adjustment brings notable gains precisely when homophily shift is severe, thereby strengthening the empirical validity of our claims.
> >
> > | Method | RU-DE | RU-FR | RU-CN | RU-JP | RU-US |
> > |-|-|-|-|-|--|
> > | ASN | 0.4053 | 0.3861 | 0.3423 | 0.3486 | 0.3662 |
> > | StruRW| 0.4235 | 0.4342 | 0.3959 | 0.3775 | 0.3822 |
> > | PairAlign | 0.4479 | 0.4728 | 0.3886 | 0.4206 | 0.4138 |
> > | HGDA | 0.5191 | 0.5253 | 0.4990 | 0.5274 | 0.5233 |
> > | our model | 0.5407 | 0.5429 | 0.5262 | 0.5396 | 0.5421 |
> >
> > **Q1.** I do not have a clear understanding of the specific theoretical contributions of this paper. I hope the authors can clarify the purpose and significance of Section 3.1 during the rebuttal stage. I will reconsider the paper and adjust my score accordingly.
> >
> > **A1.**
> >
> > We appreciate the reviewer’s request for clarification. Below, we explain the theoretical purpose and significance of Section 3.1 and how it directly motivates the design of our PSAHS framework.
> >
> > **(1) Motivation: adjusting graph structure mitigates node homophily shift**
> >
> > Our starting observation is that modifying the graph structure can effectively alter node homophily, and therefore mitigate homophily shift between domains. Guided by this insight, we adjust the graph structures in both domains to obtain $\tilde{A}_S$ and $\tilde{A}_T$. We then consider a GNN classifier trained on the adjusted source graph $\tilde{A}_S$ and evaluated on the adjusted target graph $\tilde{A}_T$.
> >
> > To understand *how* the graph structures should be specifically adjusted, Section 3 derives **Theorem 3.1**, which provides an upper bound on the target-domain error of this fitted GNN classifier.
> >
> > **(2) Interpretation of Theorem 3.1 and how it guides the design of PSAHS**
> >
> > Theorem 3.1 decomposes the target-domain error into three components. Each term offers direct and actionable guidance for how the source graph and target graph should be modified. This makes Section 3.1 the theoretical foundation for our algorithmic design.
> >
> > **Term 1 — source-domain error computed on the adjusted source graph $\tilde{A}_S$**
> >
> > This error can be reduced by **increasing the homophily ratios of low-homophily source nodes**. Accordingly, the source graph should be modified by strengthening homophilous edges and weakening heterophilous ones for low-homophily nodes, which motivates the design in Section 4.1.
> >
> > **Term 2 — Wasserstein distance between node-homophily distributions of the two adjusted graphs.**
> >
> > This indicates that to further reduce target-domain error, we should **adjust the target graph so that the distribution of homophily ratios matches that of the adjusted source graph**.
> >
> > **Term 3 — cross-domain discrepancy of GNN representations obtained from the two adjusted graphs**
> >
> > This motivates the inclusion of a **representation alignment module** to reduce cross-domain representation discrepancy.
> >
> > Taken together, these three terms give a precise blueprint for algorithm design and correspond **exactly** to the three modules of our PSAHS framework in Sections 4.1–4.3.
> >
> > **(3) Theoretical guarantees that PSAHS follows the direction suggested by Theorem 3.1**
> >
> > Section 4 not only introduces the PSAHS method but also provides theoretical results showing that the graph adjustments indeed reduce the error terms in Theorem 3.1—thereby realizing the guidance implied by the bound.
> >
> > * **Theorem 4.1** proves that the module in Section 4.1 increases the homophily ratios of low-homophily source nodes, achieving the desired form of source-graph adjustment dictated by Term 1 of Theorem 3.1.
> >
> > * **Theorem 4.2** shows that the target graph adjustment module in Section 4.2 reduces the Wasserstein distance between the node-homophily distributions, directly addressing Term 2 of Theorem 3.1.
> >
> > Thus, the theoretical contributions in Section 3.1 serve as the conceptual and mathematical basis for the PSAHS algorithm, while Section 4 provides guarantees that each module reduces the corresponding component of the derived target-domain error bound.

---

### Official Review · Reviewer_tN3o · 2025-11-02

**Soundness:** 3
**Presentation:** 3
**Contribution:** 2
**Rating:** 6
**Confidence:** 5

**Summary:**

The paper studies graph domain adaptation (GDA) under node homophily shift—the mismatch between the distributions of node-level homophily across source and target graphs. It proposes a SOTA gda architecture and their theory and observation get supported by previous research.

**Strengths:**

1. The manuscript is clearly written and well organized; notation is introduced cleanly, and figures/algorithms materially aid readability.

2. The evaluation spans a reasonably comprehensive set of datasets (synthetic and real-world) and shows consistent, meaningful gains over competitive baselines; ablations support the design choices.

3. The problem about "Node homophily shift between graph domain" is interesting and focused by many related works. I think this is a core problem of GDA.

**Weaknesses:**

1. The choice of hyperparameter seems to be central to both guarantees and performance; guidance is limited. Please include sensitivity analysis.

2. Repeated rewiring can be costly on large graphs. The paper would benefit from complexity analysis (per iteration) and wall-clock/runtime vs. accuracy plots.

3. Please add some necessary assumption in the main theorem to clarify scope.

**Questions:**

Listed in Weaknesses

---

> ### Author Response · Authors · 2025-11-28
>
> **W1.** The choice of hyperparameter seems to be central to both guarantees and performance; guidance is limited. Please include sensitivity analysis.
>
> **AW1.**
>
> To clarify the choice of hyperparameters in PSAHS, we provide detailed guidance for both the homophily threshold $h$ and the node-specific parameter $\alpha_u$, together with the sensitivity analysis that appears in the paper.
>
> **(1) Sensitivity and selection of the homophily threshold $h$**
>
> Figure 6 in Appendix D.4 provides a comprehensive sensitivity analysis of the homophily threshold $h$ across four representative tasks: B1→B2, B2→B1, Brazil→Europe, and Europe→Brazil. As shown in the four subfigures, when $h$ varies from $0.5$ to $1.0$:
>
> * each performance curve presents a **coherent and interpretable trend**, such as a monotonic increase or an increase-then-decrease pattern,
>
> * no irregular oscillations are observed, and
>
> * each task exhibits a **unique and globally stable optimal value** of $h$.
>
> These observations indicate that PSAHS is stable with respect to $h$, and that selecting $h$ via validation (as is standard in domain adaptation) does **not** introduce practical tuning difficulties.
>
> **(2) Deterministic node-specific parameter $\alpha_u$**
>
> The node-specific parameter $\alpha_u$ is **not a tunable hyperparameter**, but rather a **deterministically defined quantity** depending solely on the node homophily $h(u)$ and the selected threshold $h$:
>   $$
> \alpha_u = \frac{h - h(u)}{1 - h \cdot h(u)}.
>   $$
>
> This formulation reflects the intuition that nodes whose current homophily $h(u)$ deviates more from the target threshold $h$ should undergo larger structural adjustments. Furthermore, the choice is directly guided by **Theorem 4.2**, which shows that applying this $\alpha_u$ reduces the homophily shift more effectively than no adjustment.
>
> Appendix D.1 provides the specific $\alpha_u$ values used in practice for both domains.
> In particular:
>
> * For each low-homophily **source** node $u$, we follow Theorem 4.2 and compute
>     $$
>   \alpha_u = \frac{h - h_S(u)}{1 - h \cdot h_S(u)}.
>     $$
>
> * For each low-homophily **target** node $v$, since the true target labels are unavailable, we estimate its homophily using the pseudo-label-based approximation $\widehat{h}_T(v)$, resulting in
>   $$
>   \alpha_v = \frac{h - \widehat{h}_T(v)}{1 - h \cdot \widehat{h}_T(v)}.
>   $$
>
> Thus, all $\alpha_u$ values are **explicitly computed** and **require no parameter tuning**.
>
> **Conclusion**
>
> Both hyperparameters in PSAHS are well-justified and stable:
>
> * the homophily threshold $h$ admits clear sensitivity trends, intuitive default values, and standard validation-based selection;
>
> * the node-specific parameter $\alpha_u$ is defined in closed form, guided by theory, and involves **no tuning at all**.
>
>
> **W2.** Repeated rewiring can be costly on large graphs. The paper would benefit from complexity analysis (per iteration) and wall-clock/runtime vs. accuracy plots.
>
> **AW2.** We first provide the complexity analysis per iteration and then present runtime–accuracy results to demonstrate the scalability of our method.
>
> **(1) Complexity Analysis**
>
> Let
>
> * $N_S$, $N_T$: the number of nodes in the source and target domains,
>
> * $|\mathcal{E}_S|$ and $|\mathcal{E}_T|$: the number of edges in the source and target domains,
>
> * $\theta_T$: the ratio of reliable nodes (i.e., nodes whose GNN and MLP label predictions are aligned) among all nodes in the target domain,
>
> * $L$, $F$: the number of GNN layers and the dimension of GNN representations.
>
> In summary, the computational complexity of **one-time source graph adjustment** is
> $$
> O(N_S + |\mathcal{E}_S|),
> $$
> while the complexity of **one iteration of target graph adjustment** is
> $$
> O(LF^2 N_T + LF|\mathcal{E}_T| + (\theta_T N_T)\log (\theta_T N_T)).
> $$
>
> Below we detail how these complexities are obtained.
>
> **(a) Complexity of one-time source graph adjustment**
>
> In our method PSAHS, the source domain is processed as follows:
>
> 1. **Compute homophily and identify low-homophily nodes.**
>    We calculate the homophily ratios for all nodes and identify the low-homophily nodes, which has complexity
>    $$
>    O(N_S).
>    $$
>
> 2. **Revise all existing edges.**
>    We revise all existing edges in the source graph, which has complexity
>  $$
>    O(|\mathcal{E}_S|).
>   $$
>
> 3. **Add new same-label neighbors for low-homophily nodes.**
>    For each low-homophily node $u$, we randomly choose $d_u (1 - h_S(u))$ nodes with the same label as $u$ as new neighbors and then add new edges. This step has complexity
>   $$
>    O \Bigl( \sum_{u \in \mathcal{V}\_S} d_u (1 - h_S(u)) \Bigr)
>    = O \Bigl( \sum_{u \in \mathcal{V}\_S} d_u - \sum_{u \in \mathcal{V}\_S} d_u h_S(u) \Bigr)
>    = O(|\mathcal{E}_S| - |\mathcal{E}_S|h_S).
>   $$
>
> Therefore, the overall computation complexity of the source graph adjustment is
> $$
> O(N_S + |\mathcal{E}_S|).
> $$
>
> It is noteworthy that the source graph adjustment is conducted **only one time** before the target graph adjustment.

---

> > ### Author Response · Authors · 2025-11-28
> >
> > **(b) Complexity of a target-graph adjustment iteration**
> >
> > For the target domain, we do **not** rewire at every training step: we revise the graph structure **every 10 iterations**. During each target-graph structure adjustment iteration, PSAHS performs the following steps:
> >
> > 1. **Identify reliable target nodes.**
> >    We first identify which target nodes are reliable, i.e., the nodes with GNN–MLP aligned label predictions. Since this step requires computing the predicted labels from the GNN, it has complexity
> >    $$
> >    O(LN_T F^2 + L|\mathcal{E}_T|F).
> >    $$
> >
> > 2. **Compute homophily based on reliable neighbors.**
> >    We then calculate the node homophily ratio only based on reliable neighbors for each reliable target node. The complexity is
> >    $$
> >    O \Bigl( \sum_{u \text{ is reliable}} \theta_T d_u \Bigr)
> >    \approx O \Bigl( \theta_T \sum_{u \in \mathcal{V}_T} \theta_T d_u \Bigr)
> >    = O \bigl( \theta_T^2 |\mathcal{E}_T| \bigr),
> >    $$
> >    where $|\mathcal{E}_T|$ is the total number of target edges.
> >
> > 3. **Revise existing edges of reliable low-homophily nodes.**
> >    Next, we revise the edges of each reliable low-homophily target node $u$. Specifically, for node $u$, we increase the weights of edges connected with reliable same-class neighbors and decrease the weights of edges connected with reliable different-class neighbors. The adjustment of the existing target edges has complexity
> >    $$
> >    O \Bigl( \sum_{u \text{ is reliable}} d_u \Bigr)
> >    \approx O \Bigl( \theta_T \sum_{u \in \mathcal{V}_T} d_u \Bigr)
> >    = O \bigl( \theta_T |\mathcal{E}_T| \bigr).
> >    $$
> >
> > 4. **Select new neighbors via confidence ranking.**
> >    Moreover, we add $d_u (1 - h_T(u))$ new neighbors $v$ that are reliable, share the same predicted label with $u$, and have the highest confidence. This involves ranking the confidence for each class $k \in [K]$, where $K$ is the number of classes. Thus, the complexity of finding the new neighbors is
> >    $$
> >    O \Bigl( \sum_{k \in [K]} (\theta_T N_{T,k}) \log (\theta_T N_{T,k}) \Bigr)
> >    = O \bigl( (\theta_T N_{T}) \log (\theta_T N_{T}) \bigr),
> >    $$
> >    where $N_{T,k}$ is the number of nodes from class $k$.
> >
> > 5. **Add new edges.**
> >    Moreover, the complexity of adding the new edges is given by
> >    $$
> >    O \Bigl( \sum\_{u \text{ is reliable}} d_u (1 - h_T(u)) \Bigr)
> >    \approx O \Bigl( \theta_T \sum\_{u \in \mathcal{V}_T} d_u (1 - h_T(u)) \Bigr)
> >    = O \Bigl( \theta_T \sum\_{u \in \mathcal{V}_T} d_u - \sum\_{u \in \mathcal{V}_T} d_u h_T(u) \Bigr)
> >    = O \bigl( \theta_T |\mathcal{E}_T| (1 - h_T) \bigr),
> >    $$
> >    which is proportional to the number of heterogeneous edges in the target graph.
> >
> > Putting all parts together, the total computational complexity **per iteration** of revising the target graph structure is
> > $$
> > O(LF^2 N_T + LF|\mathcal{E}_T| + (\theta_T N_T)\log (\theta_T N_T)),
> > $$
> > where the constants $F$ and $L$ are typically very small, and $\theta_T < 1$.
> >
> > (2) The scalability analysis
> >
> > We further provide a scalability analysis in terms of wall-clock runtime and accuracy.
> >
> > We measure runtime per iteration on synthetic data to control the graph size, where the source homophily is fixed at $0.832$ and the target homophily is set as $0.3$. The first two columns in the table below report the graph size and the number of edges in the **original** target graph. The third column reports the **average running time** of all iterations. The accuracy–runtime curve is given by
> > (https://anonymous.4open.science/r/Scalability).
> >
> > | Graph Size | Number of Edges | Avg Time (s) | Accuracy (\%)   |
> > | ---------- | --------------- | ------------ | ---------------- |
> > | 300        | 365             | 0.02         | $82.13 \pm 0.85$ |
> > | 1500       | 9030            | 0.03         | $90.10 \pm 1.20$ |
> > | 3000       | 36104           | 0.04         | $93.31 \pm 5.44$ |
> > | 6000       | 143905          | 0.05         | $95.25 \pm 2.68$ |
> > | 9000       | 323879          | 0.07         | $94.76 \pm 7.21$ |
> > | 12000      | 899915          | 0.12         | $96.93 \pm 3.23$ |
> >
> > * **Accuracy:** The results show that accuracy increases substantially as the graph size grows from $300$ to $3000$, and then stabilizes at around $95\%$ as the size continues to increase up to $12000$.
> >
> > * **Runtime:** The average runtime across iterations increases at most linearly with the graph size and sublinearly with the number of edges, demonstrating the efficiency of our PSAHS method and aligning well with the complexity analysis.
> >
> > Therefore, the scalability analysis illustrates that our method PSAHS can achieve both high accuracy and efficiency, even when repeated rewiring is applied to larger graphs.

---

> > > ### Author Response · Authors · 2025-11-28
> > >
> > > **W3.** Please add some necessary assumption in the main theorem to clarify scope.
> > >
> > > **AW3.**
> > >
> > > To clarify the scope of our main Theorem 3.1, we explicitly introduce the assumptions required for the theoretical analysis. These assumptions include the Simplifying Graph Convolutional Networks (SGC) model, the Contextual Stochastic Block Model with structure (CSBM-S), equal-sized and disjoint near sets, and the concentrated expected loss difference. All of these assumptions are also adopted by previous theoretical works [1,2,3]. Below, we provide detailed explanations of each concept to make the scope of Theorem 3.1 precise.
> > >
> > > [1] Jiaqi Ma et al., *Subgroup generalization and fairness of graph neural networks*, NeurIPS 2021.
> > >
> > > [2] Haitao Mao et al., *Demystifying structural disparity in graph neural networks: Can one size fit all?*, NeurIPS 2023.
> > >
> > > [3] Ruiyi Fang et al., *Homophily enhanced graph domain adaptation*, ICML 2025.
> > >
> > > **1. SGC Model**
> > >
> > > The SGC model is one of the most commonly used GNN base classifiers and serves as the primary modeling assumption for Theorem 3.1. Specifically, SGC utilizes a one-hop mean aggregation operator followed by an MLP feature transformation. This simplification enables a tractable theoretical analysis while preserving the core behavior of message-passing GNNs.
> > >
> > > **2. CSBM-S Model**
> > >
> > > Unlike the standard Contextual Stochastic Block Model (CSBM), which assumes that all nodes follow either purely homophilic or purely heterophilic interaction patterns, the CSBM-S (CSBM with structure) model allows **more diverse and realistic** structural patterns with **varying degrees of node-level homophily**.
> > > This extension makes CSBM-S more suitable for analyzing graphs where homophily ratios $h(u)$ differ across nodes—a key property relevant to the structure-adjustment mechanism in PSAHS. The formal definition and discussion of CSBM-S appear in **Section B** and **Assumption C.1** of the appendix.
> > >
> > > **3. Equal-sized and Disjoint Near Sets**
> > >
> > > To relate the aggregated representations of source and target nodes in the theoretical bound, we introduce the assumption that all target nodes can be partitioned into **equally sized and mutually disjoint near sets**, each associated with a source node.
> > >
> > > Formally, we define the source–target representation distance as
> > > $$
> > > \epsilon := \max_{j \in V_T} \min_{i \in V_S} \|f_i(X, A) - f_j(X, A)\|_2.
> > > $$
> > >
> > > For each $i \in V_S$, the near set of $i$ is
> > > $$
> > > V_T^{(i)} := \{ j \in V_T \mid \|f_i(X, A) - f_j(X, A)\|_2 \le \epsilon \}.
> > > $$
> > >
> > > These near sets satisfy
> > > $$
> > > V_T = \bigcup_{i \in V_S} V_T^{(i)}.
> > > $$
> > >
> > > The assumption further imposes that all sets $V_T^{(i)}$ are **disjoint** and **equal in size**, which ensures that each target node is closest to exactly one source node in the representation space while being sufficiently distant from source nodes outside its corresponding set.
> > > This condition is required to control the pairwise representation discrepancy within the generalization bound. Details are provided in **Assumption C.3**.
> > >
> > > **4. Concentrated Expected Loss Difference**
> > >
> > > Finally, Theorem 3.1 relies on the assumption that the expected margin loss on the target graph
> > > $\mathcal{R}_T^{\gamma/4}$ is **not significantly larger** than the expected margin loss on the source graph
> > > $\mathcal{R}_S^{\gamma/2}$ as the source graph size $n_S$ increases.
> > >
> > > This *concentrated expected loss difference* ensures that the margin-based risks between the two domains do not deviate drastically under moderate domain shift. This standard assumption ensures that the theoretical upper bound remains meaningful. The full statement is presented in **Assumption C.4**.
> > >
> > > **Integration into the manuscript**
> > >
> > > We have integrated all the above assumptions into the revised manuscript so that the scope and applicability of Theorem 3.1 are now clearly stated and aligned with existing theoretical work.

---

### Meta-Review · Area_Chair_QYpR · 2026-01-08

**Summary:**

This submission studies node homophily shift in unsupervised graph domain adaptation and proposes PSAHS, a progressive pipeline that (i) adjusts the source graph to increase homophily for low-homophily nodes, (ii) iteratively refines the target graph using “reliable” pseudo-labels from GNN–MLP agreement, and (iii) performs domain-adversarial representation alignment. Reviewers agree that the paper targets an important phenomenon and contains a non-trivial theoretical framing and a reasonably broad experimental suite. However, the overall reception is mixed: one reviewer is above threshold, while multiple reviewers remain below threshold due to concerns about novelty differentiation, practicality/scalability, and robustness risks from pseudo-label-driven structural edits. While the rebuttal addresses many concrete points (especially those raised by reviewer fChT), I believe the remaining concerns from reviewers c2oE and F3uM are still substantial enough to recommend rejection.

**Reviewer Concerns:**

Complexity and scalability: The rebuttal provides a per-module complexity breakdown and additional runtime scaling evidence, which directly responds to the “missing complexity” critique.

Ablation/justification for reliable-node selection: The authors clarify that reliability uses deterministic GNN–MLP hard-label agreement (no additional confidence threshold), and they provide comparisons to alternative pseudo-labeling strategies to argue reduced error amplification.

Baseline-behavior explanation: The rebuttal offers a narrative to explain observed baseline behaviors under controlled homophily shifts, which improves interpretability.

Alignment novelty: The authors clarify that DANN is a modular instantiation rather than a core novelty claim, which partially resolves the “alignment component is standard” concern.

**Reviewer Scores:**

Despite the above improvements, I believe two below threshold reviewers may remain unconvinced on issues that go beyond “missing details”:

Novelty differentiation remains borderline (F3uM)
The rebuttal strengthens positioning against closely related structure/attribute-graph approaches and adds comparisons to simpler alternatives, but the core mechanism structure refinement plus pseudo-labeling with alignment may still read as an incremental combination of known ingredients. For a top-tier acceptance bar, it is not fully clear that PSAHS establishes a sharply distinct conceptual advance over recent graph-structure-learning or homophily-aware GDA lines.

Practicality and deployment risk under pseudo-label-driven edits (c2oE, F3uM)
The rebuttal argues that conservative “reliable nodes” mitigate error propagation and reports controlled experiments, but the failure mode remains structurally high-stakes: once wrong edges/weights are introduced, the method can create self-reinforcing bias. The added analyses are helpful, yet they may not fully eliminate concerns about robustness on hard, noisy, large-scale, and truly heterogeneous target graphs, especially where reliable-node coverage is limited or systematically biased.

Scalability evidence is still not fully convincing for really large graphs (c2oE)
Complexity analysis and synthetic scaling results help, but the key skepticism is practical applicability when graph sizes and domain shifts are extreme. c2oE’s rejection appears rooted in whether iterative densification/rewiring and repeated inference steps remain feasible and stable in real deployments; rebuttal evidence may still be viewed as insufficiently decisive.

Given that two reviewers (c2oE and F3uM) are likely to remain below the acceptance threshold even after rebuttal, I do not think the paper clears the acceptance bar.

---

### Decision · Program_Chairs · 2026-01-26

Reject